# Reconstruction Outcomes Look Similar but Processes Differ: Improving Context Consistency and Coverage in Graph Masked Auto-Encoder

**Geng Tang** [1]  **Keyu Liu** [1]  **Xibei Yang** [1]  **Yuhua Qian** [2]

## Abstract

Graph Masked Auto-Encoder (GMAE) has emerged as a prevalent self-supervised paradigm, showing superior performance in graph learning. However, existing methods mainly emphasize reconstruction outcomes and give limited specification to how neighborhood context is used for reconstruction. Our experimental investigation shows that, even when reconstruction outcomes are similar, the ways of using neighborhood context differ substantially, resulting in performance shifts. To address this issue, we propose Consistency- and Coverage-aware Graph Masked Auto-Encoder (C2-GMAE), which encourages more consistent use of neighborhood context and promotes broader training coverage in the graph. Specifically, C2-GMAE leverages positional encoding as an observable structural reference, introduces density-partitioned masking to improve coverage across regions, and amplifies heterophilic edges to reduce the attenuation of discriminative relational information during reconstruction. Extensive experiments on multiple benchmarks demonstrate that C2-GMAE improves downstream performance against GMAE baselines.

## 1. Introduction

Graph Masked Auto-Encoder (GMAE) has emerged as a prevalent technique for self-supervised learning on graph-structured data (Hou et al., 2022; Hu et al., 2020b; Li et al., 2023a; Tan et al., 2023). Following a corruption–reconstruction framework, it corrupts a portion of the original graph and reconstructs the masked content from

the remaining context (He et al., 2022). This design effectively captures local structural information (Tu et al., 2023), enabling the learning of generalizable representations that succeed in various tasks like node classification (Kipf, 2016; Lu et al., 2025; 2024; Xu et al., 2018b) and graph prediction (Xu et al., 2018a; Yang et al., 2022; You et al., 2021).

Recent GMAE studies have advanced the framework through designing diverse corruption strategies (Liu et al., 2024; Shi et al., 2023; Li et al., 2023b; Wang et al., 2024; Liu et al., 2025) and proposing informative reconstruction targets (Zheng et al., 2025; Zhang et al., 2025; Zhao et al., 2024; Li et al., 2023a). However, existing methods primarily concentrate on the reconstruction outcome, giving limited attention to how neighborhood context is utilized for reconstruction. Related studies (Zhu et al., 2025; Van Assel et al., 2026) have also suggested that reconstruction quality alone may not fully explain representation quality. Our focus is complementary: in masked graph reconstruction, different ways of weighting neighborhood context can yield similar reconstruction outcomes and may be associated with downstream performance shifts. We refer to this context-weighting behavior as a key part of the reconstruction process. Here, reconstruction context denotes the neighborhood information involved in reconstructing a masked target node. Since neighboring nodes are aggregated jointly, we analyze reconstruction context through the attention-based weighting over neighboring context edges, with emphasis on which edges receive higher weights.

This ambiguity in reconstruction-context usage raises two concerns. As illustrated by the toy example in Figure 1, matched reconstruction outcomes may correspond to different context usage. First, the model's use of reconstruction context varies across random seeds or masking patterns even when reconstruction outcomes are similar (Figure 1(a–b)). This suggests that reconstruction loss alone may leave multiple context-weighting patterns indistinguishable, and some of them may be less useful for downstream representation learning. Second, this ambiguity is not uniform across the graph because local reconstruction support varies by region (Figure 1(c)). In regions with strong local support, many context edges provide coherent cues for reconstruction, so reconstruction loss provides limited preference over context

---

[1]School of Computer, Jiangsu University of Science and Technology, Zhenjiang, Jiangsu, China [2]Institute of Big Data Science and Industry, Shanxi University, Taiyuan, Shanxi, China. Correspondence to: Xibei Yang <jsjxy_yxb@just.edu.cn>.

*Proceedings of the 43rd International Conference on Machine Learning*, Seoul, South Korea. PMLR 306, 2026. Copyright 2026 by the author(s).

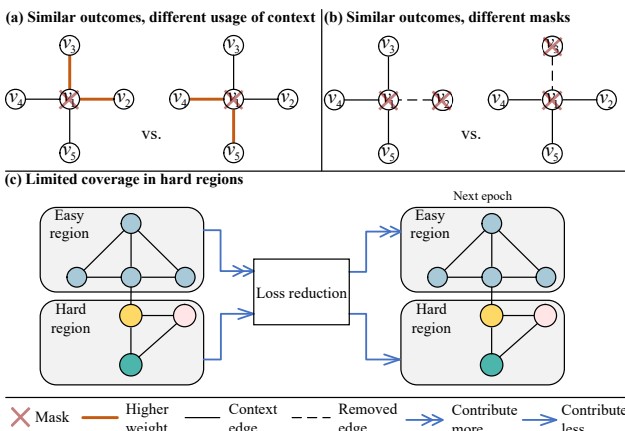

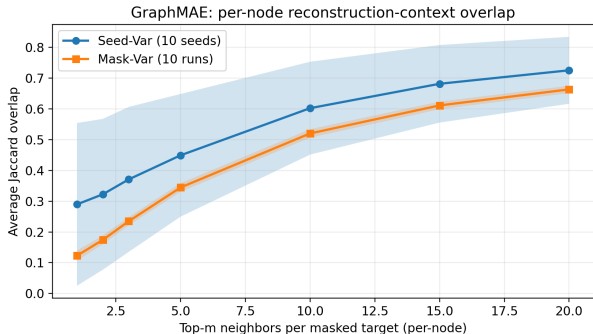

*(a)* Context overlap across seeds and masks.

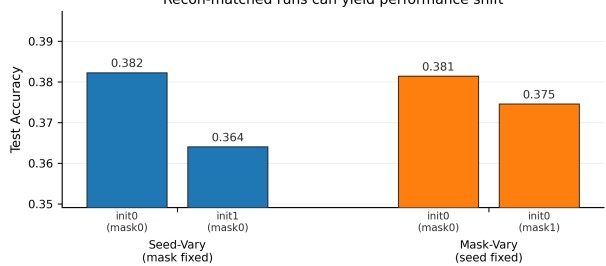

*Figure 1.* **Toy example of reconstruction-context ambiguity in GMAEs.** (a) Similar reconstruction outcomes may correspond to different high-weight context edges. (b) Different masking patterns may activate different available context edges. (c) Easy regions with strong local support may dominate loss reduction, while hard regions with weak or mixed local support receive less effective training.

*(b)* Downstream performance shift under recon-matched checkpoints.

*Figure 2.* **Controlled study on GraphMAE.** (a) Reconstruction context is measured by the top-$m$ decoder-attention neighbors of each masked target. The x-axis denotes $m$, and the y-axis denotes the average Jaccard overlap of selected neighbor sets across runs. Although larger $m$ naturally leads to higher overlap, the overlap remains incomplete. (b) Reconstruction-matched checkpoints still exhibit downstream accuracy shifts.

usage and these regions may dominate training. In regions with weak or mixed local support, the available information is less coherent, providing insufficient guidance on how to assign weights to context edges. This challenge is particularly pronounced in role-mixed neighborhoods, where nearby nodes may play different structural roles or provide mixed feature evidence, so informative cross-type relations can be attenuated by reconstruction-driven averaging.

To further examine this issue, we conduct a controlled study on GraphMAE (Hou et al., 2022), as shown in Figure 2. We select reconstruction-matched checkpoints with similar reconstruction losses and probe the reconstruction-context weighting of each masked target node using the top-$m$ neighbors with the highest decoder attention. Figure 2(a) shows that the high-weight neighbor sets only partially overlap across different training seeds or masking patterns, indicating that similar reconstruction outcomes can correspond to different context-weighting patterns. Figure 2(b) further shows noticeable downstream accuracy differences among these reconstruction-matched checkpoints. These observations motivate us to consider the reconstruction process, especially context weighting, as an additional dimension for improving GMAEs. Detailed settings and protocols are provided in Appendix A.

In this work, we aim to improve the reconstruction process by regularizing how the model weights reconstruction context when reconstructing masked targets in GMAE. Implementing this idea raises two practical challenges. First, reconstruction loss supervises the final prediction of masked nodes, but provides only indirect guidance on how attention weights should be distributed over neighboring context

edges. As a result, different edge-weighting patterns may achieve similar reconstruction outcomes. Therefore, a structural signal tied to graph edges is needed to anchor context weighting without relying on labels. Relative positional distance naturally provides such a signal because it is fixed before training, reflects graph topology, and can be injected into attention scores. Second, effective training requires sufficient coverage across regions with different local reconstruction support. Uniform masking can be dominated by easier regions where nearby nodes provide coherent reconstruction cues, leaving weakly supported or mixed regions less sufficiently trained.

In response to these challenges, we propose Consistency- and Coverage-aware Graph Masked Auto-Encoder (C2-GMAE). For the first challenge, C2-GMAE uses relative positional distances as an observable structural reference to regularize edge-level context weighting. For the second challenge, C2-GMAE introduces density-partitioned masking to allocate masking budgets across regions with different local reconstruction support. It further applies heterophilic-edge amplification to preserve informative cross-type evidence

during corruption, attention scoring, and the objective. The contributions of our paper are as follows:

1. We diagnose reconstruction-context consistency as a missing dimension in GMAEs using a controlled, reconstruction-matched analysis, and show that similar reconstruction outcomes can correspond to different attention-based context-weighting patterns.

2. We propose C2-GMAE, a consistency- and coverage-aware GMAE framework that anchors edge scoring with relative positional distances, allocates masking budgets via local-support density partitions, and amplifies heterophilic edges in both corruption and objectives.

3. We show that C2-GMAE consistently outperforms strong baselines on node- and graph-level benchmarks, and improves reconstruction-context consistency as measured by our proposed diagnostics.

## 2. Related Work

### 2.1. Masked Generative Self-Supervised Learning on Graphs

Masked generative pretraining has become a mainstream paradigm for self-supervised graph representation learning, where a model reconstructs masked parts of the input graph from the remaining context. Recent advances improve masked graph auto-encoders from multiple aspects, including stronger decoders and reconstruction targets, as well as better masking strategies and training schemes. Graph-MAE2 (Hou et al., 2023) strengthens masked pretraining by refining the decoding stage, whereas MaskGAE (Li et al., 2023a) studies masked modeling for graph auto-encoders and underscores the impact of masking choices. Masked auto-encoding has also been extended to heterogeneous graphs. More recent work treats masking itself as a key design knob, including bandwidth-style masking (Bandana) (Zhao et al., 2024), structure-guided target selection (StructMAE) (Liu et al., 2024), and curriculum scheduling of masking difficulty (Cur-MGAE) (Li et al., 2025). Recent studies also discuss the limitation of using reconstruction quality as the sole proxy for representation quality, for example from frequency-based reconstruction-representation misalignment (Zhu et al., 2025) or from comparisons between input-space reconstruction and latent-space prediction (Van Assel et al., 2026). Different from efforts that mainly improve reconstruction targets or masking heuristics, our work focuses on the reconstruction process, especially how neighborhood context is weighted during masked reconstruction, and introduces consistency and coverage mechanisms to regularize this process.

### 2.2. Graph Positional and Structural Encoding

Positional and structural encodings are widely used to complement graph attention or transformer models with global or relative structural cues. Recent work has emphasized stability and transferability in graph positional encodings (Huang et al., 2023; Kanatsoulis et al., 2025), and distance-based structural encodings for capturing multi-scale relations (Luo et al., 2024). In our setting, relative positional distances serve as an observable structural reference during masked reconstruction; we use them to anchor edge scoring and regularize context weighting, rather than to propose a new positional encoding scheme.

### 2.3. Learning under Heterophily and Heterophily-Aware Self-Supervision

Under heterophily, aggregating neighbor messages can blur discriminative signals, and recent analyses clarify when message passing helps or fails as homophily varies (Luan et al., 2023). Heterophily-aware self-supervision has therefore explored contrastive objectives paired with filter- or spectral-based view generation (Yang & Mirzasoleiman, 2023; Chen et al., 2024). Rather than proposing a new backbone, we inject heterophily awareness into masked reconstruction by upweighting heterophilic evidence in corruption, attention scoring, and the training loss. In our unsupervised setting, heterophilic edges are identified by low feature similarity over observed edges, as described in Section 4.2.

## 3. Preliminaries

### 3.1. Graph Notation

We consider an undirected graph $G = (V, E, X)$ with $N = |V|$ nodes. $X \in \mathbb{R}^{N \times F}$ denotes node features. We define the adjacency matrix $A \in \{0,1\}^{N \times N}$ by $A_{ij} = 1$ if $(v_i, v_j) \in E$ and $A_{ij} = 0$ otherwise. Let $\tilde{A} = A + I$ denote the adjacency with self-loops and let $\tilde{D}$ be its degree matrix, i.e., $\tilde{D}_{ii} = \sum_j \tilde{A}_{ij}$. We use the symmetrically normalized adjacency $\hat{A} = \tilde{D}^{-\frac{1}{2}} \tilde{A} \tilde{D}^{-\frac{1}{2}}$. For convenience, we also define the corresponding normalized Laplacian operator $L = I - \hat{A}$.

### 3.2. Laplacian Eigenvectors and Relative Positional Encoding

We assume a precomputed Laplacian eigenvector matrix $U \in \mathbb{R}^{N \times K_u}$, where the $i$-th row $u_i$ is the truncated Laplacian eigenvector coordinate of node $v_i$ provided by dataset preprocessing. We compute positional information in a relative form on an edge set $E_{\text{pe}}$ formed by adding self-loops to $E$. For each $(v_i, v_j) \in E_{\text{pe}}$, we define the relative positional distance:

$$PE_{ij} = \|u_i - u_j\|_2. \tag{1}$$

The eigenvectors are computed once before training and reused throughout all epochs. This preprocessing follows common practice in positional graph learning, but it may become a bottleneck on very large graphs; this limitation is discussed in Section 6.

## 4. Method

In this section, we propose C2-GMAE, a graph masked auto-encoder that emphasizes consistency in how neighborhood context is weighted and coverage in training. C2-GMAE is built on four components: relative positional encoding as an observable structural reference, density-partitioned masking to diversify masked targets across local-support regions, heterophilic-edge amplification to preserve heterophilic relational evidence during reconstruction, and a joint objective that combines feature reconstruction with a positional objective. Figure 3 shows an overview of the framework.

### 4.1. Positional Encoding as an Observable Structural Reference

To regularize how the model weights neighborhood context, we use an observable structural reference at the edge level. Specifically, we adopt relative positional encoding derived from Laplacian eigenvectors and an edge-wise positional regression objective, following prior masked graph modeling works(Park et al., 2022; Liu et al., 2025). In our setting, this reference serves a simple role: it anchors edge-level scoring to a fixed structural signal, thereby encouraging more consistent use of neighborhood context under random masking.

**Relative Positional Encoding.** The relative positional distance $PE_{ij}$ on $E_{\text{pe}}$ is computed from the Laplacian eigenvectors $U$ as defined in Section 3.2. This edge-wise distance is used as the positional signal throughout the framework.

**Injecting $PE$ into Attention.** An edge encoder $\text{PEG}(\cdot)$ maps the scalar $PE_{ij}$ to a head-wise bias $b_{ij} \in \mathbb{R}^{K_h}$, which is added to the pre-softmax attention score on edge $(i,j)$:

$$b_{ij} = \text{PEG}(PE_{ij}). \tag{2}$$

The bias $b_{ij}$ is added to the pre-softmax attention score, so relative positional distances directly affect the attention distribution over neighboring context edges.

### 4.2. Density-Partitioned Masking and Corruption

This subsection follows the left part of Figure 3. We first sample masked nodes by density partitions, and then corrupt both features and positional encoding. The heterophily-dependent emphasis applied at this stage corresponds to

Noise amplification (①) and PE amplification (②) in Figure 3(d).

**Density Score.** We define density as the average feature similarity between a node and its 1-hop and 2-hop neighbors, which measures local reconstruction support. A higher density score indicates that nearby nodes provide more coherent evidence for masked-node reconstruction, while a lower score indicates weaker or more mixed local support. Let $\tilde{x}_i = x_i/\|x_i\|_2$. Let $\mathcal{N}_1(i)$ be the 1-hop neighbors of $v_i$ and $\mathcal{N}_2(i)$ be the 2-hop neighbors. Define $\mathcal{N}_{1:2}(i) = \big(\mathcal{N}_1(i) \cup \mathcal{N}_2(i)\big) \setminus \{i\}$. We compute:

$$d_i = \begin{cases} \frac{1}{|\mathcal{N}_{1:2}(i)|} \sum_{j \in \mathcal{N}_{1:2}(i)} \tilde{x}_i^\top \tilde{x}_j, & |\mathcal{N}_{1:2}(i)| > 0, \\ 0, & \text{otherwise.} \end{cases} \tag{3}$$

**Partition and Equal Sampling.** We sort nodes by $d_i$ in descending order and split them into $B$ disjoint partitions $V_1, \ldots, V_B$ of similar size. Given a mask ratio $r$, set $m = \lfloor rN/B \rfloor$ and randomly sample the same number of nodes from each partition. Let $S_b$ denote the sampled set from $V_b$ with $|S_b| = m$. The masked node set is formed as:

$$S = \bigcup_{b=1}^{B} S_b. \tag{4}$$

This procedure is a stratified masking strategy that exposes each masking step to different local-support levels.

**Feature Corruption.** Based on $S$, we construct $X^{\text{masked}}$ by masking nodes in $S$ with a learnable mask token and optional feature replacement: a subset of masked nodes is zeroed and filled with the learnable mask token, while the remaining masked nodes are replaced by random node features. Nodes outside $S$ keep their original features.

**Heterophilic Edges Indicator.** Since labels are unavailable during pretraining, we identify heterophilic edges in an unsupervised way based on feature similarity over observed edges. We compute cosine similarity on $E_{\text{pe}}$: $\text{sim}_{ij} = \tilde{x}_i^\top \tilde{x}_j$. Let $q_p$ be the value such that a fraction $p$ of non-self-loop edges have similarity no larger than $q_p$. The indicator is defined as:

$$H_{ij} = \begin{cases} 1, & \text{sim}_{ij} \leq q_p, \ i \neq j, \\ 0, & \text{otherwise,} \end{cases} \tag{5}$$

and $H_{ii} = 0$ is used for self-loops. Edges with $H_{ij} = 1$ are treated as identified heterophilic edges in the following components. This indicator is computed from observed features and fixed during training.

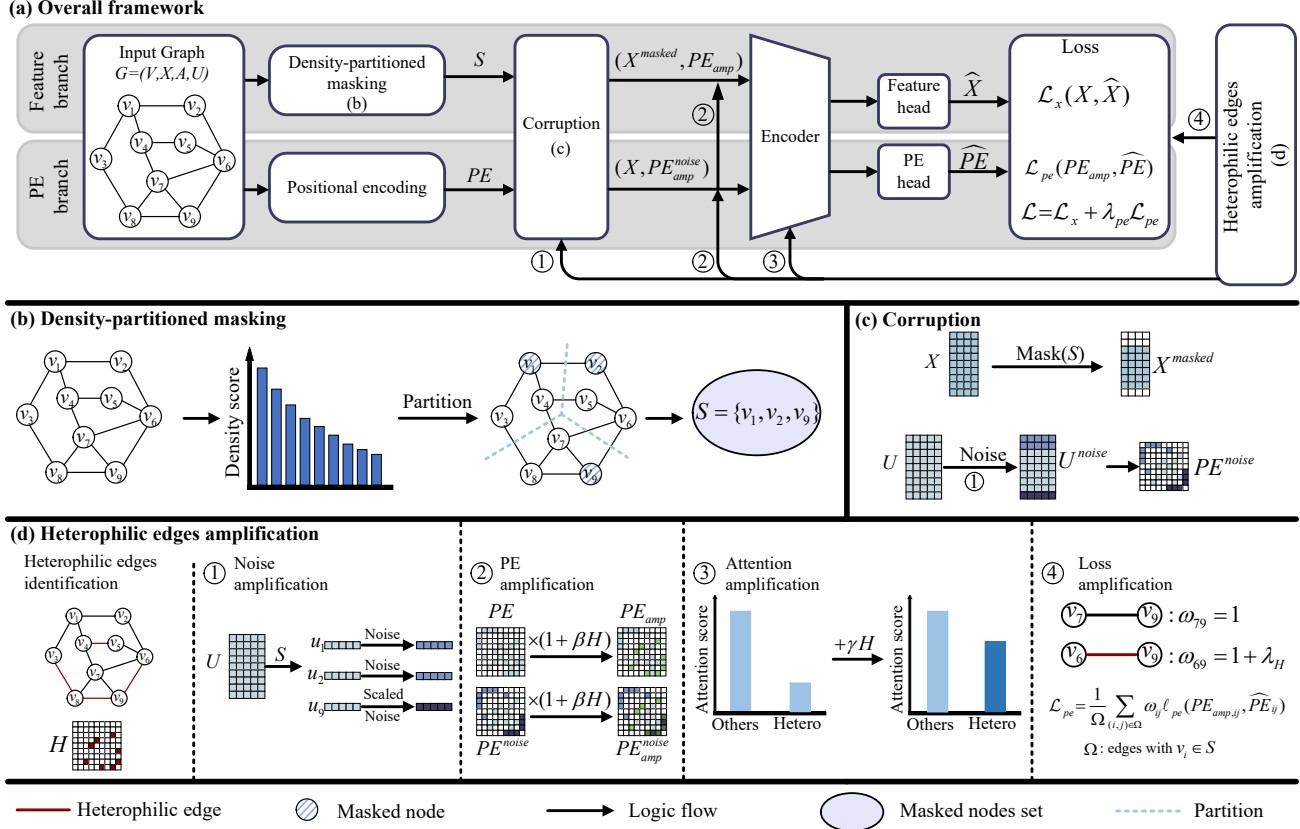

**Figure 3.** **Overview of C2-GMAE.** Starting from an input graph $G = (V, X, A, U)$, density-partitioned masking module samples a masked node set $S$. Corruption module then builds two inputs for a shared attention-based encoder: the Feature branch uses $(X^{\text{masked}}, PE_{amp})$ to reconstruct node features via the Feature head, while the PE branch uses $(X, PE_{amp}^{\text{noise}})$ to predict edge-wise positional distances via the PE head. The training objective combines feature reconstruction loss $\mathcal{L}_x$ and positional loss $\mathcal{L}_{pe}$. Panel (a) is the overall framework of C2-GMAE. Panel (b) illustrates density-based partitioning module and stratified sampling of masked nodes. Panel (c) shows feature masking and positional perturbation that produce $X^{\text{masked}}$ and $PE^{\text{noise}}$. Panel (d) summarizes heterophilic edges amplification module: heterophilic edges are identified by $H$, and emphasis is applied at four places: ① scaling positional noise for masked nodes, ② scaling positional distances to obtain $PE_{amp}$ and $PE_{amp}^{\text{noise}}$, ③ adding a heterophily gate to pre-softmax attention scores, and ④ weighting heterophilic edges in computing loss.

**Positional Corruption with Noise Amplification (①).** We add uniform noise to Laplacian eigenvectors of masked nodes:

$$\epsilon_i \sim \text{Unif}(-\mu, \mu)^{K_u}, \qquad v_i \in S. \tag{6}$$

where $\mu$ is set to 0.001 or 0.01 for different datasets.

Let $\deg(i)$ be the degree of $v_i$ on the self-loop removed and symmetrized version of $E_{\text{pe}}$, and let $\deg_H(i)$ be the number of incident edges with $H_{ij} = 1$ on the same edge set. The noise magnitude for masked nodes is scaled by:

$$\epsilon_i \leftarrow \left(1 + \kappa \cdot \frac{\deg_H(i)}{\deg(i) + 10^{-8}}\right)\epsilon_i. \tag{7}$$

Then:

$$u_i^{\text{noise}} = \begin{cases} u_i + \epsilon_i, & v_i \in S, \\ u_i, & v_i \notin S. \end{cases} \tag{8}$$

This increases positional perturbation around masked nodes that are incident to more identified heterophilic edges, mak-

ing the PE branch more sensitive to locally mixed regions during reconstruction.

**PE Amplification (②).** We recompute positional distances on $E_{\text{pe}}$: $PE_{ij}^{\text{noise}} = \|u_i^{\text{noise}} - u_j^{\text{noise}}\|_2$, and amplify distances on heterophilic edges:

$$\begin{aligned} PE_{\text{amp},ij} &= (1 + \beta H_{ij})\, PE_{ij}, \\ PE_{\text{amp},ij}^{\text{noise}} &= (1 + \beta H_{ij})\, PE_{ij}^{\text{noise}}. \end{aligned} \tag{9}$$

Applying the same amplification to clean and perturbed distances keeps the PE input and the regression target on a consistent scale, while increasing the contribution of identified heterophilic edges in the positional objective.

### 4.3. Encoder with Attention Amplification

A shared attention-based encoder is evaluated on two branches. The Feature branch takes $(X^{\text{masked}}, PE_{\text{amp}})$

while the PE branch takes $(X, PE_{\text{amp}}^{\text{noise}})$. The Feature head reconstructs node features, and the PE head predicts edge-wise distances from PE-branch edge-level scores.

**Attention Amplification (③).** Let $s_{ij}^{(h)}$ denote the pre-softmax attention score of head $h$ after adding the PE bias. In each attention layer, we add a heterophily-dependent gate to the positional bias:

$$s_{ij}^{(h)} \leftarrow s_{ij}^{(h)} + \gamma H_{ij}, \qquad h = 1, \ldots, K_h. \quad (10)$$

This increases the attention score of identified heterophilic edges, encouraging them to retain stronger influence in the attention distribution during reconstruction.

## 4.4. Objectives with Loss Amplification

Our final loss consists of a feature loss $\mathcal{L}_x$ and a positional loss $\mathcal{L}_{pe}$.

**Feature Loss.** Denote the reconstructed feature of node $v_i$ as $\hat{x}_i$. Following (Zheng et al., 2025), we instantiate $\ell(\cdot, \cdot)$ as a Scaled Cosine Error (SCE):

$$\cos(\hat{x}, x) = \frac{\hat{x}^\top x}{\|\hat{x}\|_2 \|x\|_2 + \epsilon}, \qquad \ell(\hat{x}, x) = \big(1 - \cos(\hat{x}, x)\big)^\alpha. \quad (11)$$

Masked reconstruction is computed by:

$$\mathcal{L}_{\text{mask}} = \frac{1}{|S|} \sum_{v_i \in S} \ell(\hat{x}_i, x_i). \quad (12)$$

A residual target on unmasked nodes is formed by one-step smoothing and subtraction (Zheng et al., 2025):

$$X^D = (I - \hat{A})X. \quad (13)$$

Here $x_i^D$ denotes the $i$-th row of $X^D$, and $\hat{x}_i^D$ denotes the corresponding decoder prediction for the residual target. We compute:

$$\mathcal{L}_{\text{unmask}} = \frac{1}{|V \setminus S|} \sum_{v_i \notin S} \ell(\hat{x}_i^D, x_i^D). \quad (14)$$

They are combined as:

$$\mathcal{L}_x = \mathcal{L}_{\text{mask}} + \lambda_d \mathcal{L}_{\text{unmask}}. \quad (15)$$

**Positional Loss with Loss Amplification (④).** We focus positional supervision on edges around masked nodes. We remove self-loops and symmetrize $E_{\text{pe}}$ to obtain an undirected edge set $\tilde{E}_{\text{pe}}$. We define:

$$\Omega = \{(i, j) \in \tilde{E}_{\text{pe}} \mid v_i \in S\}. \quad (16)$$

Let $\widehat{PE}_{ij}$ be the PE head prediction from PE-branch edge scores. We regress it to the clean amplified target $PE_{\text{amp}, ij}$ using Smooth-$\ell_1$. We upweight heterophilic edges by:

$$\omega_{ij} = 1 + \lambda_H H_{ij}, \quad (17)$$

and compute:

$$\mathcal{L}_{pe} = \frac{1}{|\Omega|} \sum_{(i,j) \in \Omega} \omega_{ij} \cdot \text{Smooth-}\ell_1\big(\widehat{PE}_{ij}, PE_{\text{amp}, ij}\big). \quad (18)$$

Therefore, the final loss function of C2-GMAE is:

$$\mathcal{L} = \mathcal{L}_x + \lambda_{pe} \mathcal{L}_{pe}. \quad (19)$$

# 5. Experiments

## 5.1. Node Classification

**Datasets.** We evaluate C2-GMAE on 8 graphs: Facebook (Rozemberczki et al., 2021), WikiCS (Mernyei & Cangea, 2020), BlogCatalog (Meng et al., 2019), Chameleon, Squirrel, Actor (Pei et al., 2020), arXiv-year (Hu et al., 2020a), and Penn94 (Traud et al., 2012). Facebook and WikiCS are homophilic graphs, while the others are heterophilic graphs. We further include arXiv-year and Penn94 as large-scale benchmarks.

**Baselines and Settings.** We compare C2-GMAE with a wide range of graph self-supervised baselines, which can be grouped into two categories: **(1) Contrastive learning methods:** DGI (Veličković et al., 2018), BGRL (Tan et al., 2023), MVGRL (Hassani & Khasahmadi, 2020), CCA-SSG (Zhang et al., 2021), and Sp$^2$GCL (Bo et al., 2023). **(2) Graph auto-encoders:** VGAE (Kipf & Welling, 2016), GraphMAE (Hou et al., 2022), GraphMAE2 (Hou et al., 2023), MaskGAE (Li et al., 2023a), S2GAE (Tan et al., 2023), AUG-MAE (Wang et al., 2024), TEDMAE (Zhang et al., 2025), and GraphPAE (Liu et al., 2025).

For all methods, we use the same encoder backbone: a GAT with 4 heads and 1024 hidden units. The number of layers is searched in $\{2, 3\}$. For efficiency and scalability, we use two-layer MLPs with ReLU activation as both the feature decoder and the position decoder. Following the standard evaluation protocol, we freeze the pretrained encoder and extract node representations. We then train a linear classifier with labeled data for node classification. For all methods, we use the Adam optimizer and run 10 times on each graph.

**Results.** Table 1 summarizes the node classification results. C2-GMAE ranks first on all eight datasets. On the heterophilic graphs, the improvements are more visible, especially on Actor and Penn94 where C2-GMAE gives the highest accuracy with a clear margin. On the two homophilic graphs (Facebook and WikiCS), C2-GMAE also

*Table 1.* Node classification results of different graph self-supervised learning methods, mean accuracy (%) ± standard deviation. (Bold denotes the best result)

| Dataset | Small Graphs | | | | | | Large Graphs | |
|---|---|---|---|---|---|---|---|---|
| | BlogCatalog | Chameleon | Squirrel | Actor | Facebook | WikiCS | arXiv-year | Penn94 |
| DGI | 72.07±0.16 | 43.83±0.14 | 34.56±0.10 | 27.98±0.09 | 86.37±0.70 | 57.89±1.93 | - | - |
| BGRL | 79.74±0.46 | 61.24±1.07 | 43.24±0.52 | 26.61±0.57 | 89.71±0.35 | 79.02±0.13 | 41.43±0.04 | 63.31±0.49 |
| MVGRL | 63.24±0.94 | 73.19±0.42 | 60.09±0.44 | 34.64±0.20 | 80.29±0.20 | 76.94±0.60 | - | - |
| CCA-SSG | 74.00±0.28 | 75.00±0.75 | 61.58±1.98 | 27.79±0.58 | 89.45±0.60 | 78.85±0.32 | 40.78±0.01 | 62.63±0.20 |
| Sp$^2$GCL | 72.73±0.46 | 78.88±1.04 | 62.61±0.87 | 34.70±0.92 | 86.75±0.45 | 77.25±0.50 | 39.09±0.02 | 68.80±0.45 |
| VGAE | 60.47±1.84 | 62.32±1.90 | 42.50±1.35 | 31.57±0.75 | 68.56±0.60 | 64.26±0.47 | 36.39±0.21 | 55.31±0.28 |
| GraphMAE | 79.90±1.13 | 79.50±0.57 | 61.13±0.60 | 32.15±1.33 | 89.54±0.36 | 78.94±0.48 | 40.30±0.04 | 67.97±0.21 |
| GraphMAE2 | 77.34±0.12 | 79.13±0.19 | 70.27±0.88 | 34.48±0.26 | 88.49±0.43 | 78.84±0.44 | 38.97±0.03 | 67.86±0.42 |
| MaskGAE | 73.10±0.08 | 74.50±0.87 | 68.53±0.44 | 33.44±0.34 | 85.3±0.22 | 76.96±0.37 | 40.59±0.04 | 63.84±0.03 |
| S2GAE | 75.76±0.43 | 60.24±0.37 | 68.60±0.56 | 26.17±0.38 | 91.50±0.20 | 74.16±0.41 | 40.32±0.12 | 70.24±0.09 |
| AUG-MAE | 82.03±0.69 | 70.10±1.88 | 62.57±0.67 | 33.42±0.38 | 86.65±0.30 | 78.09±0.40 | 37.10±0.13 | 69.90±0.43 |
| TEDMAE | 72.47±1.01 | 64.28±1.13 | 68.76±0.62 | 29.61±0.50 | 91.20±0.20 | 79.38±0.25 | 39.89±0.08 | 66.41±0.30 |
| GraphPAE | 85.76±1.22 | 80.51±1.25 | 72.05±1.40 | 38.55±1.35 | 91.46±0.23 | 79.32±0.29 | 41.85±0.04 | 71.79±0.37 |
| **OURS** | **90.38±0.33** | **82.42±0.71** | **74.80±0.49** | **42.59±1.12** | **91.57±0.16** | **79.59±0.15** | **42.16±0.04** | **74.17±0.25** |

achieves the best performance. Taken together, C2-GMAE achieves the best results among the compared methods on the evaluated node classification benchmarks, with particularly visible gains on several heterogeneous graph datasets.

## 5.2. Graph Prediction

**Datasets.** For graph-level tasks, we evaluate C2-GMAE on 6 OGB datasets (Hu et al., 2020a), including 3 graph regression tasks and 3 graph classification tasks. For all datasets, we use public splits for a fair comparison. We use RMSE and ROC-AUC as the evaluation metrics for regression and classification, respectively.

**Baselines and Settings.** We compare with 5 contrastive learning methods, i.e., InfoGraph (Sun et al., 2019), GraphCL (You et al., 2020), MVGRL (Hassani & Khasahmadi, 2020), JOAO (You et al., 2021), and Sp$^2$GCL (Bo et al., 2023), and 5 graph auto-encoders, i.e., GraphMAE (Hou et al., 2022), GraphMAE2 (Hou et al., 2023), StructMAE (Liu et al., 2024), AUG-MAE (Wang et al., 2024), and GraphPAE (Liu et al., 2025). We use a two-layer GatedGCN as the encoder. We adopt two-layer MLPs as feature and position decoders. For evaluation, we freeze the encoder to output node representations and input them into pooling functions for graph representations. We then train a linear predictor on the graph representations for downstream evaluation. We use Adam optimizer and report the metrics with mean results and standard deviation of 10 seeds.

**Results.** Table 2 reports the results on OGB graph regression and classification benchmarks. C2-GMAE performs best on all six datasets. For the three regression tasks, it achieves the lowest RMSE on molesol, mollipo, and mol-

freesolv. For the three classification tasks, it achieves the highest ROC-AUC on molbace, molclintox, and moltox21. These results show that the proposed design is also effective for graph-level representation learning, beyond node classification.

## 5.3. Ablation Study

We conduct ablation studies on four datasets (BlogCatalog, Chameleon, Squirrel, and Actor). We follow the same training and linear evaluation protocol as in the main experiments and report mean accuracy (%) ± standard deviation over 10 runs.

**Settings and Variants.** We start from the full model **Full** (C2-GMAE) and consider two ablated variants. **Variant A** disables density-partitioned masking (DP) and uses uniform masking. **Variant B** disables heterophilic edges amplification (HA), while keeping other components unchanged.

**Analysis.** Table 3 evaluates the two coverage-related components. Replacing density-partitioned masking with uniform masking (Variant A) reduces performance on most datasets, especially Chameleon and Actor, indicating that stratifying masked nodes by local reconstruction support is useful. Disabling heterophilic-edge amplification (Variant B) also causes consistent drops, suggesting that heterophilic edges provide useful reconstruction evidence in these benchmarks. The role of positional anchoring is further analyzed through reconstruction-context diagnostics in Appendix B.

**Where Heterophilic Edges Amplification Matters.** We further decompose heterophilic edges amplification mod-

*Table 2.* Graph regression and classification results of different graph self-supervised learning on OGB datasets. (Bold indicates the best result. ↓ means lower the better and ↑ means higher the better.)

| Task | Regression (Metric: RMSE ↓) | | | Classification (Metric: ROC-AUC% ↑) | | |
|---|---|---|---|---|---|---|
| Dataset | molesol | mollipo | molfreesolv | molbace | molclintox | moltox21 |
| InfoGraph | 1.344±0.178 | 1.005±0.023 | 10.005±8.147 | 73.64±3.64 | 64.50±5.32 | 69.74±0.57 |
| GraphCL | 1.272±0.089 | 0.910±0.016 | 7.679±2.748 | 73.32±2.70 | 74.92±4.42 | 72.40±1.07 |
| MVGRL | 1.433±0.145 | 0.962±0.036 | 9.024±1.982 | 74.88±1.43 | 73.84±2.75 | 70.48±0.83 |
| JOAO | 1.285±0.121 | 0.865±0.032 | 5.131±0.782 | 74.43±1.94 | 71.28±4.12 | 71.38±0.92 |
| Sp$^2$GCL | 1.235±0.119 | 0.835±0.026 | 4.144±0.573 | 78.76±1.43 | 80.88±3.86 | 73.06±0.75 |
| GraphMAE | 1.050±0.034 | 0.850±0.022 | 2.740±0.233 | 79.14±1.31 | 80.56±5.55 | 73.84±0.58 |
| GraphMAE2 | 1.225±0.081 | 0.885±0.019 | 2.913±0.293 | 80.74±1.53 | 75.75±3.65 | 72.93±0.69 |
| StructMAE | 1.499±0.043 | 1.089±0.002 | 2.568±0.262 | 77.75±0.42 | 79.42±4.56 | 71.13±0.61 |
| AUG-MAE | 1.248±0.026 | 0.917±0.013 | 2.395±0.158 | 78.54±2.49 | 82.66±1.98 | 74.33±0.07 |
| GraphPAE | 1.015±0.045 | 0.810±0.018 | 2.058±0.188 | 81.11±1.24 | 82.69±3.39 | 74.46±0.54 |
| **OURS** | **0.996±0.052** | **0.805±0.016** | **2.026±0.143** | **81.97±3.16** | **82.70±2.32** | **74.61±0.45** |

*Table 3.* Main ablations on four datasets (mean accuracy (%) ± std). DP: density-partitioned masking; PE: relative positional encoding as a structural reference; HA: heterophilic edges amplification.

| Variant | DP | PE | HA | BlogCatalog | Chameleon | Squirrel | Actor |
|---|---|---|---|---|---|---|---|
| Full | ✓ | ✓ | ✓ | **90.38±0.33** | **82.42±0.71** | **74.80±0.49** | **42.59±1.12** |
| A | ✗ | ✓ | ✓ | 90±1.12 | 80.59±0.92 | 74.68±0.61 | 40.67±1.85 |
| B | ✓ | ✓ | ✗ | 89.22±1.99 | 80.90±1.08 | 74.06±1.13 | 41.53±0.86 |

ule into four components defined in Sec. 4.2–4.4: Noise amplification ($\kappa$, Eq. (7)), PE amplification ($\beta$, Eq. (9)), Attention amplification ($\gamma$, Eq. (10)), and Loss amplification ($\lambda_H$, Eq. (17)). We consider four variants, each disabling one component.

**Analysis.** Table 4 decomposes heterophilic-edge amplification into four sites. PE amplification and attention amplification usually contribute the most, which matches their direct roles in changing the positional reference and edge-level context weights. Noise amplification improves performance by strengthening perturbations around masked nodes incident to more heterophilic edges. Loss amplification gives smaller but mostly positive gains, indicating that heterophilic edges remain useful in the positional objective.

### 5.4. Parameter Analysis

We analyze the sensitivity of the four coefficients in heterophilic edges amplification module on Actor and Chameleon. The four coefficients correspond to noise amplification ($\kappa$), PE amplification ($\beta$), attention amplification ($\gamma$), and loss amplification ($\lambda_H$). For each coefficient, we vary it from 0.1 to 1.0 while fixing the other three. We report mean accuracy over 10 runs. For readability, we plot the mean curves without error bars in Figure 4 and Figure 5.

**Actor.** Figure 4 shows different preferred ranges across components on Actor. For PE amplification ($\beta$), the best

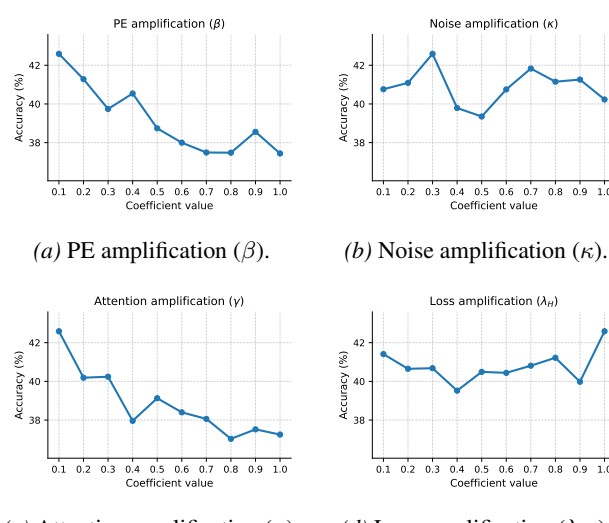

*(a)* PE amplification ($\beta$).  *(b)* Noise amplification ($\kappa$).

*(c)* Attention amplification ($\gamma$). *(d)* Loss amplification ($\lambda_H$).

*Figure 4.* Parameter sensitivity on Actor.

performance appears at a small value (0.1), and larger $\beta$ gradually reduces accuracy. Noise amplification ($\kappa$) shows a mild peak around 0.3, while values away from this range are slightly worse. For Attention amplification ($\gamma$), the best result is achieved at a small value (0.1), and increasing $\gamma$ tends to hurt performance. In contrast, Loss amplification ($\lambda_H$) prefers a larger value in our search, and 1.0 gives the best accuracy.

*Table 4.* Ablations on different components of *Heterophilic edges amplification* (mean accuracy (%) ± std). $\kappa$: noise amplification; $\beta$: PE amplification; $\gamma$: attention amplification; $\lambda_H$: loss amplification.

| Variant | $\kappa$ | $\beta$ | $\gamma$ | $\lambda_H$ | BlogCatalog | Chameleon | Squirrel | Actor |
|---------|----------|---------|----------|-------------|-------------|-----------|----------|-------|
| Full | ✓ | ✓ | ✓ | ✓ | **90.38±0.33** | **82.42±0.71** | **74.80±0.49** | **42.59±1.12** |
| D | ✓ | ✗ | ✓ | ✓ | 87.85±1.05 | 81.04±2.14 | 74.25±1.48 | 40.33±2.64 |
| E | ✗ | ✓ | ✓ | ✓ | 89.78±0.93 | 81.86±1.22 | 74.48±0.86 | 40.05±2.89 |
| F | ✓ | ✓ | ✗ | ✓ | 88.44±0.52 | 81.90±1.27 | 74.30±1.01 | 39.72±1.20 |
| G | ✓ | ✓ | ✓ | ✗ | 88.23±0.50 | 81.83±0.96 | 74.40±0.57 | 41.09±2.75 |

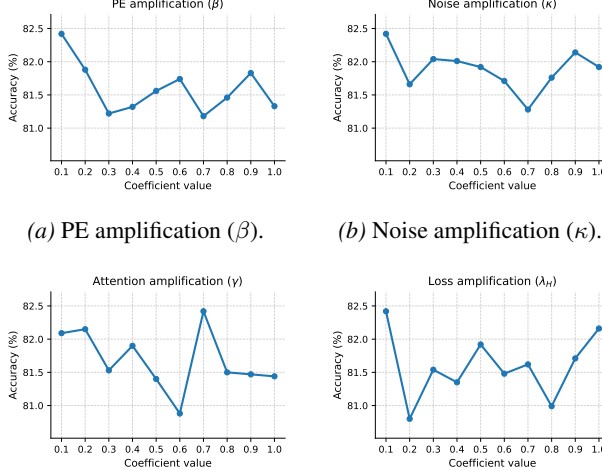

*(a)* PE amplification ($\beta$).

*(b)* Noise amplification ($\kappa$).

*(c)* Attention amplification ($\gamma$).

*(d)* Loss amplification ($\lambda_H$).

*Figure 5.* Parameter sensitivity on Chameleon.

**Chameleon.** Figure 5 shows that Chameleon prefers small values for PE amplification ($\beta$) and Noise amplification ($\kappa$), with the best performance at 0.1. Attention amplification ($\gamma$) behaves differently on this dataset: the best result is obtained at a moderate value (0.7), while smaller or larger values are slightly worse. For Loss amplification ($\lambda_H$), the best accuracy also appears at 0.1, and increasing $\lambda_H$ leads to a small drop.

**Practical Hyperparameter Selection.** In practice, we first set the number of density partitions $B$ according to graph size. We choose $B$ so that each partition contains roughly 50 nodes, which keeps each density group sufficiently populated while preserving a fine-grained stratification of local reconstruction support. For very small graph-level datasets, a smaller $B$ is used to avoid overly sparse partitions. After fixing $B$, we tune the standard training hyperparameters, including the mask ratio $r$, dropout rate $dp$, and the loss weights $\lambda_{pe}$ and $\lambda_d$. The HA coefficients $\kappa$, $\beta$, $\gamma$, and $\lambda_H$ are then searched within small ranges. Among them, $\gamma$ is usually the most dataset-sensitive term because it directly changes the pre-softmax attention scores, while $\kappa$ and $\lambda_H$ tend to affect performance more smoothly.

**Summary.** Across both datasets, PE amplification works best with mild scaling, while overly large values reduce accuracy. Attention amplification is the most dataset-sensitive: Actor prefers a small gate, whereas Chameleon peaks at a moderate value, reflecting different neighbor-importance patterns across graphs. Noise amplification and loss amplification change performance more smoothly and are best treated as tuning knobs, with their preferred ranges varying by dataset.

## 6. Limitations

C2-GMAE has several limitations. First, it relies on pre-computed truncated Laplacian eigenvectors. Although this is a one-time offline step and the resulting eigenvectors are reused during training, the preprocessing can still become a bottleneck on massive graphs. Second, the effect of heterophilic-edge amplification depends on graph characteristics. It is designed to preserve informative heterophilic evidence in role-mixed neighborhoods, but its gain may become smaller on strongly homophilic graphs, and its behavior under more extreme heterophilic settings may vary. Third, C2-GMAE introduces additional hyperparameters and edge-wise computations. This increases model-selection cost and brings extra time and memory overhead compared with simpler GMAE baselines.

## 7. Conclusion

In this paper, we study GMAEs from the perspective of the reconstruction process, with a focus on reconstruction-context weighting. Through reconstruction-matched diagnostics, we show that similar reconstruction outcomes can correspond to different attention-based context-weighting patterns and downstream performance shifts. We propose C2-GMAE to regularize edge-level context weighting and broaden effective reconstruction coverage. The framework combines relative positional anchoring, density-partitioned masking, and heterophilic-edge amplification. Experiments on node-level and graph-level benchmarks, together with ablation and diagnostic analyses, support the effectiveness of the proposed design.

## Impact Statement

This paper presents work whose goal is to advance the field of machine learning. There are many potential societal consequences of our work, none of which we feel must be specifically highlighted here.

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

# A. Context-Variation Study

This section details the controlled study reported in the main text (Figure 2). We investigate how a GMAE weights reconstruction context when reconstruction outcomes are similar. Unless otherwise stated, the main diagnostics in this section are conducted on the Squirrel dataset. Additional Chameleon diagnostics are reported in Appendix B.4.

## A.1. Reconstruction-Matched Checkpoints

We compare reconstruction-matched checkpoints to control for reconstruction quality. For each run, we save the checkpoint whose training reconstruction loss is closest to a target value (e.g., 0.70). These checkpoints have comparable reconstruction outcomes, which allows the diagnostic to compare attention-based context-weighting patterns under a matched reconstruction level.

## A.2. Measuring Reconstruction Context: Per-Node Top-$m$ vs. Global Top-$k$

Decoder-attention overlap is used as a practical diagnostic proxy for reconstruction-context weighting. It does not provide a complete causal attribution of reconstruction behavior, but it reflects which context edges receive larger decoder attention scores and is suitable for comparing context-weighting stability across controlled runs.

We quantify reconstruction context in two ways. The per-node Top-$m$ overlap in Figure 2 and the global Top-$k$ overlap used in the claim verification serve different purposes. The per-node Top-$m$ overlap is a local diagnostic for visualizing whether the high-attention neighbors of each masked target are stable. The global Top-$k$ overlap is a graph-level summary that gives one scalar for comparing different runs and variants. We use the former for intuition and visualization, and the latter for compact systematic verification.

**Per-Node Top-$m$ Context (Figure 2).** For each masked target node $i$, we rank its neighbors by decoder attention score and take the top-$m$ neighbors, denoted by $\mathcal{N}_m(i)$. Given two runs (or two probing masks) $A$ and $B$, we compute per-node Jaccard overlap and average it over masked targets:

$$J_m(A, B) = \frac{1}{|S|} \sum_{i \in S} \frac{|\mathcal{N}_m^A(i) \cap \mathcal{N}_m^B(i)|}{|\mathcal{N}_m^A(i) \cup \mathcal{N}_m^B(i)|}, \tag{20}$$

where $S$ is the masked target set in the probing pass. Figure 2 reports the mean overlap as a function of $m$.

**Global Top-$k$ Context (Claims and Tables).** To quantify variation at a graph level, we record decoder attention scores on all context edges during probing, rank these edges globally, and select the top-$k$ edges with the highest scores. We denote this set by $\mathcal{E}_k$. The overlap between two runs $A$ and $B$ is measured by the Jaccard similarity:

$$J_k(A, B) = \frac{|\mathcal{E}_k^A \cap \mathcal{E}_k^B|}{|\mathcal{E}_k^A \cup \mathcal{E}_k^B|}. \tag{21}$$

A lower $J_k$ indicates less overlap among high-attention edges under the diagnostic metric, even if the final reconstruction loss is matched. Appendix B also uses $J_k$.

## A.3. Two Sources of Context Randomness: Seed-Var and Mask-Var

We consider two controlled settings.

**Seed-var (mask fixed).** We train models with different random seeds. During probing, we use the same fixed mask pattern for all runs. We report the mean and standard deviation of overlap across training seeds.

**Mask-var (checkpoint fixed).** We fix a trained checkpoint (encoder/decoder weights are held constant) and only vary the random mask pattern in the probing pass. We report overlap across these probing masks.

## A.4. Performance Shift under Context Variation

To examine downstream differences under context variation, we use a standard linear probing protocol. For each reconstruction-matched checkpoint, we freeze the encoder and train a linear classifier for node classification. Since

reconstruction quality is matched, downstream accuracy differences are compared under a controlled reconstruction condition.

## B. Claim-Based Verification (Claim0–Claim2)

We organize the evidence into three claims. Unless noted otherwise, all measurements follow the Global Top-$k$ overlap $J_k$ defined in Eq. (21).

### B.1. Claim0: Similar Reconstruction Outcomes Can Correspond to Different Reconstruction Contexts

Claim0 states that runs with matched reconstruction loss can still exhibit different high-weight reconstruction contexts. This is observed in both Seed-var and Mask-var settings: the selected high-attention edge sets are not perfectly stable, even when reconstruction outcomes are similar. Figure 2 illustrates this phenomenon at the per-node level using $J_m$ in Eq. (20). Tables 5 and 6 further report the corresponding global Top-$k$ overlap $J_k$ under seed variation and mask variation.

### B.2. Claim1: Our Method Makes Reconstruction-Context Weighting More Consistent

Claim1 checks whether C2-GMAE encourages the model to weight reconstruction context more consistently. We compare the full model with two variants:

- **Noanchor**: removes the positional encoding.

- **Nohint**: removes the heterophilic-edge amplification.

The results are reported in Table 5 and Table 6.

#### B.2.1. SEED-VAR RESULTS (MASK FIXED)

Table 5 reports Seed-var results with $k \in \{100, 500, 1000\}$. The results show that the full C2-GMAE model has higher overlap in global top-$k$ edges across training seeds compared to the baselines.

*Table 5.* Seed-var (mask fixed): Mean$\pm$std of Global Top-$k$ Jaccard $J_k$ over training seeds.

| Fixed mask | $k$ | Full | Noanchor | Nohint |
|---|---|---|---|---|
| 0 | 100 | 0.5306$\pm$0.0419 | 0.4234$\pm$0.0515 | 0.3956$\pm$0.0475 |
| | 500 | 0.5147$\pm$0.0362 | 0.4776$\pm$0.0374 | 0.4772$\pm$0.0417 |
| | 1000 | 0.4864$\pm$0.0341 | 0.4529$\pm$0.0387 | 0.4338$\pm$0.0384 |
| 1 | 100 | 0.5418$\pm$0.0502 | 0.4874$\pm$0.0322 | 0.4497$\pm$0.0343 |
| | 500 | 0.5260$\pm$0.0364 | 0.4804$\pm$0.0337 | 0.4730$\pm$0.0322 |
| | 1000 | 0.4949$\pm$0.0295 | 0.4522$\pm$0.0311 | 0.4489$\pm$0.0327 |
| 2 | 100 | 0.5773$\pm$0.0326 | 0.5178$\pm$0.0389 | 0.4575$\pm$0.0425 |
| | 500 | 0.5147$\pm$0.0390 | 0.4772$\pm$0.0334 | 0.4774$\pm$0.0288 |
| | 1000 | 0.4967$\pm$0.0428 | 0.4589$\pm$0.0363 | 0.4499$\pm$0.0319 |

#### B.2.2. MASK-VAR RESULTS (CHECKPOINT FIXED, $k = 100$ TO $1000$)

Table 6 reports Mask-var results. Here we fix a trained checkpoint and vary only the probing masks. The full model maintains reasonable consistency, while Noanchor drops sharply, indicating strong dependence on the specific random mask view without the structural reference.

### B.3. Claim2: Our Method Improves Training Coverage Over Density Regions

Claim2 checks whether training coverage is more balanced across regions with different densities. We group nodes into $B$ density bins and count how many Global Top-$k$ edges belong to each bin based on the masked target node endpoint. This gives a normalized bin budget distribution $p$. We report three metrics: entropy of $p$, bin-level standard deviation of

*Table 6.* Mask-var (checkpoint fixed): Mean±std of Global Top-$k$ Jaccard $J_k$ across probing masks.

| $k$ | Full | Noanchor | Nohint |
|---|---|---|---|
| 100 | 0.184±0.029 | 0.054±0.065 | 0.178±0.022 |
| 200 | 0.190±0.021 | 0.057±0.068 | 0.180±0.013 |
| 300 | 0.186±0.013 | 0.057±0.067 | 0.177±0.015 |
| 400 | 0.183±0.012 | 0.060±0.071 | 0.176±0.011 |
| 500 | 0.182±0.010 | 0.060±0.071 | 0.174±0.010 |
| 600 | 0.182±0.010 | 0.060±0.072 | 0.175±0.009 |
| 700 | 0.185±0.006 | 0.060±0.072 | 0.178±0.006 |
| 800 | 0.187±0.002 | 0.060±0.072 | 0.180±0.006 |
| 900 | 0.188±0.003 | 0.061±0.073 | 0.183±0.008 |
| 1000 | 0.187±0.005 | 0.062±0.075 | 0.186±0.009 |

$p$ (Bin-Std), and the total budget count (BudgetCnt). Higher entropy and lower Bin-Std indicate more balanced coverage across density bins.

We evaluate two bin assignments:

- **Real bin:** the original node-to-bin assignment based on density.

- **Shuffled bin:** a random assignment that keeps the bin sizes but breaks the link between node density and bin index.

Table 7 reports the results on Squirrel. With the real bin assignment, the full model has higher entropy and lower Bin-Std than the random baseline, indicating a more balanced budget distribution. With the shuffled bin assignment, this trend changes, confirming that the improvement comes from density partitioning rather than an artifact of counting.

*Table 7.* Claim2 coverage metrics on Squirrel. Values are reported as mean±std across five seeds. Entropy and Bin-Std are computed on the normalized bin budget distribution $p$.

| Setting | Metric | Full | Random |
|---|---|---|---|
| Real bin | Entropy | 4.3297±0.0015 | 4.3129±0.0027 |
| | Bin-Std | 0.0073±0.0000 | 0.0076±0.0000 |
| | BudgetCnt | 114827±407 | 119120±293 |
| Shuffled bin | Entropy | 4.4528±0.0010 | 4.4590±0.0021 |
| | Bin-Std | 0.0056±0.0000 | 0.0055±0.0000 |
| | BudgetCnt | 114827±407 | 119120±293 |

## B.4. Additional Diagnostics on Chameleon

To examine whether the diagnostic conclusions are specific to Squirrel, we further conduct the Claim0/1/2 diagnostics on Chameleon. Table 8 reports the Claim0/1 context-overlap stability results, and Table 9 reports the Claim2 coverage metrics. The same overall trends are observed.

*Table 8.* Claim0/1 context-overlap stability on Chameleon. Higher is better.

| Setting | Full | Noanchor | Nohint |
|---|---|---|---|
| Seed-var, $J_k$ @100/500/1000 | 0.497 / 0.527 / 0.503 | 0.448 / 0.456 / 0.489 | 0.392 / 0.487 / 0.496 |
| Mask-var, $J_k$ @100/500/1000 | 0.199 / 0.195 / 0.195 | 0.095 / 0.094 / 0.092 | 0.192 / 0.191 / 0.192 |

The full model shows better context-overlap stability than Noanchor under both Seed-var and Mask-var. For Claim2, the full model shows slightly higher entropy and lower or comparable Bin-Std under real density bins. After bin shuffling, the difference between Full and Random largely disappears, suggesting that the coverage effect comes from meaningful density partitions rather than random grouping.

*Table 9.* Claim2 coverage metrics on Chameleon. Entropy and Bin-Std are computed on the normalized distribution over density bins. Higher entropy and lower Bin-Std indicate more balanced coverage.

| Bins | Full Entropy | Full Bin-Std | Random Entropy | Random Bin-Std |
|---|---|---|---|---|
| Real, $k = 1000$ | 3.601 | 0.0108 | 3.597 | 0.0113 |
| Real, $k = 500$ | 3.582 | 0.0118 | 3.579 | 0.0118 |
| Shuffled, $k = 1000$ | 3.673 | 0.0045 | 3.675 | 0.0041 |
| Shuffled, $k = 500$ | 3.667 | 0.0052 | 3.667 | 0.0051 |

### B.5. Comparison with GraphMAE on Context Stability

To directly compare with the motivating baseline, we evaluate attention-based reconstruction-context overlap for C2-GMAE and GraphMAE under the same Seed-var and Mask-var protocols. For Squirrel Seed-var, the C2-GMAE values are averaged over the three fixed-mask settings. Table 10 reports the comparison on Chameleon and Squirrel.

*Table 10.* Context-overlap stability comparison between C2-GMAE and GraphMAE on Chameleon and Squirrel.

| Setting | Chameleon C2-GMAE | Chameleon GraphMAE | Squirrel C2-GMAE | Squirrel GraphMAE |
|---|---|---|---|---|
| Seed-var, $k = 100$ | 0.497 | 0.497 | 0.550 | 0.491 |
| Seed-var, $k = 500$ | 0.527 | 0.479 | 0.519 | 0.348 |
| Seed-var, $k = 1000$ | 0.503 | 0.460 | 0.493 | 0.346 |
| Mask-var, $k = 100$ | 0.199 | 0.166 | 0.184 | 0.179 |
| Mask-var, $k = 500$ | 0.195 | 0.159 | 0.182 | 0.143 |
| Mask-var, $k = 1000$ | 0.195 | 0.162 | 0.187 | 0.140 |

C2-GMAE gives more stable reconstruction context than GraphMAE on both datasets. The gap is clearer when $k$ is larger.

### B.6. Summary of Verification Results

In summary, reconstruction-matched checkpoints can still exhibit substantial variation in reconstruction context (Claim0), under both seed variation and mask variation. C2-GMAE improves context consistency by anchoring edge scoring to a structural reference and strengthening heterophilic cues (Claim1), as shown in Tables 5, 6, and 8. It also promotes broader training coverage across density regions via density-partitioned masking (Claim2), as shown in Tables 7 and 9. The comparison with GraphMAE in Table 10 further supports the diagnostic results.

## C. Additional Experiments and Analysis

### C.1. Additional Homophilic Benchmarks

We further evaluate C2-GMAE on five standard homophilic benchmarks. Table 11 reports the results. C2-GMAE remains competitive on these datasets, although the gains are smaller and not uniform across all cases.

*Table 11.* Additional results on standard homophilic benchmarks.

| Dataset | GraphMAE | GraphMAE2 | GraphPAE | Ours |
|---|---|---|---|---|
| Cora | 84.20±0.40 | 84.50±0.60 | 82.22±0.60 | **85.13±0.21** |
| Citeseer | 73.40±0.40 | **73.40±0.30** | 72.30±0.58 | 72.14±0.56 |
| Pubmed | 81.10±0.40 | 81.40±0.50 | 81.40±0.25 | **82.60±0.51** |
| Photo | 93.19±0.39 | 93.14±0.56 | 91.90±0.30 | **93.86±1.24** |
| Computers | 90.02±0.24 | 88.87±0.42 | 88.85±0.30 | **90.80±1.13** |

## C.2. Effect of Heterophilic-Edge Amplification on Homophilic Graphs

Since HA is designed to preserve heterophilic evidence, its effect on homophilic graphs is expected to be smaller. Table 12 reports the ablation on Facebook and WikiCS. HA brings modest gains on both datasets, suggesting that it does not introduce noticeable noise in these evaluated homophilic graphs.

*Table 12.* Effect of heterophilic-edge amplification on homophilic graphs.

| Dataset | Full C2-GMAE | w/o HA |
|---|---|---|
| Facebook | **91.57±0.16** | 91.29±0.16 |
| WikiCS | **79.59±0.15** | 78.88±0.15 |

## C.3. Effect of Neighborhood Range in Density Estimation

We compare different neighborhood ranges for density estimation. The 1-hop score can be sensitive to immediate neighborhood noise, while 3-hop and 4-hop neighborhoods may include distant nodes less relevant to masked reconstruction. The 2-hop score gives a better balance between locality and stability. Table 13 reports the results on Actor and Chameleon.

*Table 13.* Effect of neighborhood range in density estimation.

| Hop Range | Actor | Chameleon |
|---|---|---|
| 1-hop | 39.82±1.00 | 81.62±0.83 |
| 2-hop | **42.59±1.12** | **82.42±0.71** |
| 3-hop | 41.65±1.74 | 82.16±0.68 |
| 4-hop | 41.36±1.22 | 82.05±0.92 |

## C.4. Comparison with Adaptive Heterophilic Masking

We further compare density-partitioned masking with an adaptive masking alternative that increases the masking probability for lower-density, more heterophilic nodes. Nodes are first divided into $B$ partitions by density. The $k$-th partition is assigned a weight controlled by $b$, and its masking quota is proportional to this weight. A larger $b$ assigns more masking budget to lower-density partitions, which contain nodes with weaker local reconstruction support. Table 14 shows that simply increasing the masking probability of such nodes is less effective than the proposed stratified masking strategy.

*Table 14.* Comparison with adaptive heterophilic masking.

| Dataset | Ours | $b = 2.0$ | $b = 3.0$ |
|---|---|---|---|
| Chameleon | **82.42±0.71** | 81.50±1.11 | 81.10±1.00 |
| Actor | **42.59±1.12** | 41.08±1.02 | 40.89±2.27 |

## C.5. Encoder Backbone Analysis

C2-GMAE is not restricted to GAT, but its current node-level instantiation is better matched with attention-based encoders because the method modulates edge-wise context scoring through positional bias and heterophily-aware gating. Table 15 compares GAT and GCN backbones. The gap is larger on heterophilic graphs, where fixed normalized aggregation in GCN is less suitable for the proposed edge-wise scoring mechanisms.

## C.6. Complexity and Preprocessing Cost

Let $N$ be the number of nodes, $|E|$ the number of edges, $K_u$ the positional dimension, and $L$ the number of encoder layers. The main additional preprocessing cost of C2-GMAE comes from computing truncated Laplacian eigenvectors. For sparse graphs, this step is performed once per graph with approximate cost $O(TK_u(|E| + N))$, where $T$ denotes the number of

*Table 15.* Encoder backbone analysis.

| Dataset | GAT | GCN |
|---|---|---|
| Actor | **42.59±1.12** | 31.15±0.33 |
| Chameleon | **82.42±0.71** | 52.86±1.11 |
| WikiCS | **79.59±0.15** | 77.68±0.51 |

solver iterations. The resulting $U \in \mathbb{R}^{N \times K_u}$ is stored and reused during training. This preprocessing is separate from training, but it can still dominate the total cost on very large graphs.

During training, the additional PE computation is sparse and edge-wise on $E_{\text{pe}}$. Computing or storing $PE_{ij}$, $PE_{ij}^{noise}$, and $H_{ij}$ over $E_{\text{pe}}$ introduces a constant-factor overhead over sparse message passing. The positional loss is computed only on $\Omega$, the edge subset incident to masked nodes, with cost $O(|\Omega|)$. Extra memory mainly comes from storing $U$, optional edge-wise PE or $H$ statistics, and PE-branch scores and targets on $\Omega$.

# D. Experimental Details

## D.1. Detailed Statistics

We evaluate C2-GMAE on node classification and graph prediction tasks. For all datasets, we use public splits. The statistics of node classification and graph prediction datasets are shown in Tables 16 and 17, respectively.

*Table 16.* Statistics of node classification datasets.

| Datasets | Nodes | Edges | Features | Classes | Train/Valid/Test |
|---|---|---|---|---|---|
| BlogCatalog | 5196 | 343486 | 8189 | 6 | 120/1000/1000 |
| Chameleon | 2277 | 62792 | 2325 | 5 | 1092/729/456 |
| Squirrel | 5201 | 396846 | 2089 | 5 | 2496/1664/1041 |
| Actor | 7600 | 53411 | 932 | 5 | 3648/2432/1520 |
| Facebook | 22470 | 171002 | 4714 | 4 | 2247/2247/17976 |
| WikiCS | 11701 | 297110 | 300 | 10 | 580/1769/5847 |
| arXiv-year | 169343 | 2315598 | 128 | 5 | 84671/42335/42337 |
| Penn94 | 41554 | 2724458 | 4814 | 2 | 19407/9703/9705 |

*Table 17.* Statistics of graph prediction datasets.

| Datasets | Graphs | Avg.Nodes | Avg.Edges | Classes |
|---|---|---|---|---|
| molesol | 1128 | 13.3 | 13.7 | 1 |
| mollipo | 4200 | 27.0 | 29.5 | 1 |
| molfreesolv | 642 | 8.7 | 8.4 | 1 |
| molbace | 1513 | 34.1 | 36.9 | 1 |
| molclintox | 1477 | 26.2 | 27.9 | 2 |
| moltox21 | 7831 | 18.6 | 19.3 | 12 |

## D.2. Detailed Hyperparameters

Hyperparameters for node classification and graph prediction are reported in Tables 18 and 19. Here, $r$ is the mask ratio and $dp$ denotes the dropout rate. $\lambda_{pe}$ and $\lambda_d$ are the coefficients of $\mathcal{L}_{pe}$ and $\mathcal{L}_{unmask}$, respectively. $B$ is the number of density partitions, which is mainly determined by graph size so that each partition contains roughly 50 nodes when possible. The parameters $\kappa$, $\beta$, $\gamma$, and $\lambda_H$ control heterophilic-edge amplification, where $\kappa$ controls noise amplification, $\beta$ controls PE amplification, $\gamma$ controls attention amplification, and $\lambda_H$ controls loss amplification.

*Table 18.* Hyperparameters of node classification.

| **Datasets** | $r$ | $dp$ | $\lambda_{pe}$ | $\lambda_d$ | $B$ | $\kappa$ | $\beta$ | $\gamma$ | $\lambda_H$ |
|---|---|---|---|---|---|---|---|---|---|
| BlogCatalog | 0.25 | 0.6 | 0.1 | 0.1 | 100 | 0.8 | 0.1 | 1.0 | 1.0 |
| Chameleon | 0.25 | 0.0 | 0.01 | 0.1 | 50 | 0.1 | 0.1 | 0.1 | 0.1 |
| Squirrel | 0.3 | 0.6 | 0.001 | 0.01 | 100 | 0.1 | 0.1 | 0.1 | 0.1 |
| Actor | 0.25 | 0.1 | 0.1 | 0.1 | 110 | 0.3 | 0.1 | 0.1 | 1.0 |
| Facebook | 0.25 | 0.5 | 0.1 | 0.1 | 430 | 0.1 | 0.1 | 0.1 | 0.1 |
| WikiCS | 0.2 | 0.1 | 0.1 | 0.01 | 580 | 0.5 | 0.01 | 0.01 | 0.05 |
| arXiv-year | 0.5 | 0.0 | 0.01 | 0.1 | 3400 | 0.1 | 0.1 | 0.1 | 0.1 |
| Penn94 | 0.25 | 0.0 | 0.001 | 0.1 | 800 | 0.8 | 0.1 | 1.0 | 1.0 |

*Table 19.* Hyperparameters of graph prediction.

| **Datasets** | $r$ | $dp$ | $\lambda_{pe}$ | $\lambda_d$ | $B$ | $\kappa$ | $\beta$ | $\gamma$ | $\lambda_H$ |
|---|---|---|---|---|---|---|---|---|---|
| molesol | 0.75 | 0.6 | 0.1 | 0.1 | 2 | 0.8 | 0.1 | 1.0 | 0.9 |
| mollipo | 0.25 | 0.6 | 0.001 | 0.01 | 4 | 0.8 | 0.4 | 1.0 | 0.5 |
| molfreesolv | 0.5 | 0.5 | 0.1 | 0.1 | 2 | 0.2 | 0.2 | 0.1 | 0.1 |
| molbace | 0.75 | 0.5 | 0.1 | 0.1 | 3 | 0.8 | 0.2 | 0.8 | 1.0 |
| molclintox | 0.25 | 0.6 | 0.01 | 0.01 | 3 | 0.5 | 0.4 | 0.4 | 1.0 |
| moltox21 | 0.25 | 0.0 | 0.1 | 0.1 | 2 | 0.5 | 0.1 | 0.8 | 1.0 |

## D.3. Implementation Summary

For node classification, all methods use the same GAT encoder with 4 heads and 1024 hidden units, with the number of layers searched in $\{2, 3\}$. Both feature and PE decoders are two-layer MLPs. For graph-level tasks, a two-layer GatedGCN encoder and two-layer MLP decoders are used. All reported results are averaged over 10 runs. The code will be released to support reproducibility.

