# OpenReview forum: "Reconstruction Outcomes Look Similar but Processes Differ: Improving Context Consistency and Coverage in Graph Masked Auto-Encoder"
_ICML.cc/2026/Conference — ICML 2026 regular_

### Official Review · Reviewer_g3q1 · 2026-02-21

**Soundness:** 3
**Presentation:** 2
**Significance:** 2
**Originality:** 3
**Overall Recommendation:** 5
**Confidence:** 5

**Summary:**

This paper addresses a critical issue in Graph Masked Auto-Encoders: although models may achieve similar reconstruction errors, the "neighborhood context" used to reconstruct masked nodes can vary significantly, leading to fluctuations in downstream task performance. Furthermore, regions within the graph exhibiting different local support densities can result in unbalanced training coverage. To address these issues, the authors propose C2-GMAE.

The model incorporates three core innovations: 1) utilizing relative positional encodings as an observable structural reference; 2) introducing a density-partitioned masking mechanism; and 3) amplifying the influence of heterophilic edges during both the corruption and objective computation phases to mitigate the attenuation of discriminative relational information. Experimental results demonstrate that C2-GMAE outperforms current strong baseline models across multiple node classification and graph-level prediction benchmarks.

**Compliance With Llm Reviewing Policy:**

Affirmed.

**Final Justification:**

The author's response addressed the issue I raised, and I am willing to recommend accepting this paper.

**Key Questions For Authors:**

1. The method introduces a large number of hyperparameters, and the hyperparameter settings vary significantly across different datasets (Table 10, Table 11), which poses a challenge for applying the method to new datasets.
2. What distinguishes the positional reconstruction component from GraphPAE? This requires further elaboration.
3. Is this method limited to running on a GAT encoder? How is the performance when using a GCN encoder?

**Limitations:**

Performance under extreme heterophilic graph scenarios and the impact of heterophilic edge amplification on it, the complex selection of hyperparameters, as well as the time overhead and space consumption introduced by the method.

**Strengths And Weaknesses:**

**Strengths**
1. The motivation is clearly articulated, with the authors providing concrete empirical analysis that demonstrates variations in context usage across different experimental settings.
2. The proposed density-partitioned masking mechanism effectively addresses the issue of imbalanced training coverage, while the heterophilic edges amplification strategy successfully preserves the model's discriminative power in complex, role-mixed neighborhoods.
3. Comprehensive evaluations on eight node classification benchmarks and six OGB graph-level prediction datasets consistently demonstrate state-of-the-art performance.

**Weaknesses**
1. The key concepts, specifically "Context Consistency" and "Coverage," lack detailed explanations and formal definitions. Furthermore, their connection to the proposed methodology is not clearly established.
2. The organization of the methodology section appears disordered. For example, it is unclear how $b_{ij}$ in Equation 2 participates in the subsequent training process. Moreover, $x^D_i$ in Equation 14 lacks a definition, and the justification for introducing $\mathcal{L}_{unmask}$ is not provided.
3. The experimental evaluation primarily focuses on heterophilic graphs, lacking validation on standard homophilic datasets such as Cora, Citeseer, Pubmed, Computers, and Photo.
4. The method incorporates multiple loss functions, including positional reconstruction (edge-level) and feature reconstruction (node-level). An efficiency analysis covering time complexity and memory consumption should be provided, as this is critical for scalability to large-scale datasets.
5. The ablation study presented in Table 3 is insufficient, as it lacks fine-grained ablation on individual components. Based on the current results, the method still outperforms baselines even when single components are removed, suggesting the need for more detailed analysis to verify the necessity of each module.
6. While Heterophilic Edges Amplification demonstrates advantages on heterophilic graphs (e.g., Actor, Chameleon), it is unclear whether forcibly amplifying heterophilic edges introduces additional noise in highly homophilic graphs.
7. The hyperparameter sensitivity analysis in Figures 4 and 5 indicates that performance is highly sensitive to hyperparameter choices across different datasets. Guidance on how to determine optimal parameter selections under such multi-hyperparameter settings is needed.

---

> ### Author Rebuttal · Authors · 2026-03-31
>
> Dear Reviewer g3q1:
>
> Thank you for the helpful comments. Our responses are as follows:
>
> **Response to Weakness 1:**
> “Context consistency” means using neighborhood context more consistently when reconstructing a masked node across seeds or masking patterns, instead of relying on unstable incidental choices under similar reconstruction loss. “Coverage” has two aspects: (1) at the masking stage, masked nodes should cover regions with different local support rather than concentrate on easy regions; (2) at the reconstruction stage, the model should not over-rely on easier homophilic neighbors, but should also better use informative heterophilic neighbors.
>
> **Response to Weakness 2:**
> In Eq. (2), $b_{ij}=PEG(PE_{ij})$ is the edge-wise positional bias added to the pre-softmax attention score, so it directly affects edge scoring during training. In Eq. (14), we will restate that $x_i^D$ is the residual target from Eq. (13). We will also clarify $\mathcal{L}_{unmask}$: it supervises unmasked nodes with the residual/high-pass signal, preventing the model from focusing only on easy smooth reconstruction.
>
> **Response to Weakness 3:**
> We added five homophilic benchmarks, showing that C2-GMAE also works well on homophilic graphs.
>
> | Data | GMAE | GMAE2 | GPAE | Ours |
> |---|---:|---:|---:|---:|
> | Cora | 84.20±0.40 | 84.50±0.60 | 82.22±0.60 | **85.13±0.21** |
> | Citeseer | **73.40±0.40** | **73.40±0.30** | 72.30±0.58 | 72.14±0.56 |
> | Pubmed | 81.10±0.40 | 81.40±0.50 | 81.40±0.25 | **82.60±0.51** |
> | Photo | 93.19±0.39 | 93.14±0.56 | 91.90±0.30 | **93.86±1.24** |
> | Computers | 90.02±0.24 | 88.87±0.42 | 88.85±0.30 | **90.80±1.13** |
>
> **Response to Weakness 4:**
> We agree that efficiency and scalability should be clarified more explicitly. The additional cost introduced by C2-GMAE remains sparse and edge-wise during training, so it mainly brings a constant-factor overhead over the backbone rather than changing the asymptotic order. We will add a clearer discussion in the revision; for the detailed time and memory complexity analysis, please see **Response to Question 2 of Reviewer QY8r**.
>
> **Response to Weakness 5:**
> In Table 3, removing DP causes the largest drops on Chameleon and Actor, matching its role in improving coverage on harder regions. Removing HA causes consistent drops across datasets. Table 4 shows that attention amplification and PE amplification are usually the strongest HA components; noise amplification is also important, while loss amplification is smaller but consistently positive.
>
> **Response to Weakness 6:**
> Reconstruction naturally favors neighbors that reduce loss more easily. In heterophilic graphs, the fewer homophilic neighbors are often easier to fit, so the model may overuse them and underuse the more abundant but harder heterophilic relations; this is why heterophilic evidence needs amplification. In homophilic graphs, homophilic edges already dominate, so amplifying the small number of heterophilic edges does not overturn the main pattern or add noticeable noise. For direct evidence on homophilic graphs, please also see **Response to Question 4 of Reviewer QY8r**.
>
> **Response to Weakness 7:**
> We agree that practical tuning guidance is needed, for details please see **Response to Question 1**.
>
> **Response to Question 1:**
> The four HA coefficients are usually effective in small ranges. $B$ is also not fully free, since we choose it so that each partition has roughly 50 nodes, making it largely determined by graph size. A practical strategy is: set $B$ by graph size, tune the standard training/objective parameters, then do a small-range search for HA, with attention amplification treated as the most dataset-sensitive term.
>
> **Response to Question 2:**
> The key difference is mechanism: GraphPAE corrupts node positions and learns a dedicated position path that updates positional representations, making position reconstruction a primary representation-learning target. In C2-GMAE, $PE_{ij}$ is mapped by $PEG(\cdot)$ to an edge bias for attention scoring, and the PE branch predicts edge-wise distances mainly to regularize how masked-node context edges are scored.
>
> **Response to Question 3:**
> Our framework is not limited to GAT, but its current node-level instantiation is better matched with attention-based encoders than with GCN, because our method modulates edge-wise context scoring through positional bias and heterophily-aware gating, while GCN uses fixed normalized aggregation weights. This mismatch is especially harmful on heterophilic graphs; on homophilic graphs the drop is much smaller.
>
> | Data | GAT | GCN |
> |---|---:|---:|
> | Actor | 42.59±1.12 | 31.15±0.33 |
> | Chameleon | 82.42±0.71 | 52.86±1.11 |
> | WikiCS | 79.59±0.15 | 77.68±0.51 |
>
> **Response to Limitations:**
> We agree that our manuscript lacks a limitations section. We will add it in final version, for more details please see **Response to Limitations of Reviewer E6kc**.

---

> > ### Author Rebuttal · Reviewer_g3q1 · 2026-04-01
> >
> > Having reviewed the authors' rebuttal, I have decided to raise my score.

---

> > > ### Author Response · Authors · 2026-04-07
> > >
> > > We sincerely thank you for reviewing our rebuttal and for raising your score. We are very glad that our clarification addressed your concerns.

---

### Official Review · Reviewer_Jv2E · 2026-03-01

**Soundness:** 2
**Presentation:** 1
**Significance:** 1
**Originality:** 1
**Overall Recommendation:** 2
**Confidence:** 5

**Summary:**

The paper proposes C2-GMAE, a graph masked autoencoder (GMAE) framework for graphs that improves how neighborhood context is utilized during reconstruction. It introduces two main mechanisms: density-partitioned masking and corruption and heterophilic edge amplification. The proposed framework outperforms existing GMAE baselines on both node classification and graph prediction.

**Compliance With Llm Reviewing Policy:**

Affirmed.

**Final Justification:**

The rebuttal has addressed some of my concerns, but some weaknesses still remain (see my rebuttal acknowledgement).

Based on the current situation, despite my appreciation of the authors' efforts, I believe this paper still does not reach the expected quality of the ICML conference. I will maintain my score.

**Key Questions For Authors:**

See weaknesses above.

**Limitations:**

yes

**Strengths And Weaknesses:**

Strengths:

* The proposed framework consistently outperforms the newly proposed GMAE baselines like TEDMAE and GraphPAE on both node-level and graph-level downstream tasks.

Weaknesses:

* **Writing and illustration are difficult to understand.** For instance, what exactly is "reconstruction context"? Does it refer to the masked neighborhood of a node during pre-training? Other terms like "role-mixed neighborhoods" (line 093), "reconstruction-matched checkpoints" (line 098-099), and "boundary and conflict cues" (line 077) are not explained. Fig. 1 is confusing and its caption is too brief. What do "limited coverage" and "hard regions" refer to? What is the unit and meaning of the x-axis in Fig. 2 (maybe the average number of $m$)? What do its rising curves indicate?

* **Motivation is unclear.** The training loss of an MAE depends only on the decoder's output, so why would similar reconstruction outcomes "result in performance shift" (line 042)? Varying masking patterns are a nature of GraphMAE, which employs a random mask distribution for every epoch of a single training process, so why is such "under-determination" a main problem in GMAEs? Performance variance due to random seeds is also common across many models, and attributing it to the uncertainty of masked context is not convincing.

* **Poor methodology soundness.** The paper lacks intuitive or theoretical analysis for the proposed C2-GMAE. There is almost no experiment to verify its effectiveness against the aforementioned issues of GraphMAE, making its design purpose unclear. Why is density $d_i$ of a node defined as the average inner product with its 2-hop neighbors? How is the graph split into partitions with similar size, and what is its advantage over existing graph partitioning algorithms? Additionally, what is the definition of "heterophily" used in this work? How does taking the bottom-$p$ least similar node pairs from $E_{pe}$​ capture heterophily in the graph?

* **Reproducibility is not guaranteed.**

---

> ### Author Rebuttal · Authors · 2026-03-31
>
> Dear Reviewer Jv2E:
>
> Thank you for valuable comments and questions. Our responses are as follows:
>
> **Response to Weakness 1:**
>
> We agree that several terms and figures were not clear enough in the main text, which were actually clarified in Appendix. We now clarify the main terms as follows.
> - Reconstruction context: the neighbor information used to reconstruct a masked node. In Fig. 2, we probe it by the top-$m$ neighbors with the highest decoder attention.
> - Role-mixed neighborhoods: neighborhoods with mixed homophilic and heterophilic information.
> - Reconstruction-matched checkpoints: checkpoints with very similar reconstruction loss, so the compared runs are at a similar reconstruction level.
> - Boundary/conflict cues: informative signals that may be weakened when the model mainly averages over dominant neighbors.
> - Hard regions: regions with higher heterophily and weaker local support, so they are harder to reconstruct.
> - Limited coverage: training tends to focus on easier, more homophilic regions, while harder, more heterophilic regions receive less effective training.
>
> For Fig. 2(a), the x-axis is m, and the y-axis is the average Jaccard overlap of the top-m neighbor sets across runs. The curves increase with m because larger sets overlap more easily, but the overlap remains incomplete. We will revise these terms and improve the captions of Figs. 1-2.
>
>
> **Response to Weakness 2:**
>
> Our claim is not that randomness itself is a special problem of GMAEs. Random masking and random seeds are common in many models. Our point is that, in GMAEs, reconstruction loss tells the model what to reconstruct, but does not fully control which neighbors are used or how they are weighted. So similar reconstruction loss does not always mean similar reconstruction context. In Fig. 2, we first choose checkpoints with similar reconstruction loss and then examine the top-*m* attention-based reconstruction context. We find that these contexts still overlap only partially across seeds or probing masks, and such checkpoints can still lead to different downstream accuracy.
>
> **Response to Weakness 3:**
>
> We agree that the design reason should have been made clearer. The appendix already includes claim-based verification, and we further add comparisons between C2-GMAE and GraphMAE on Chameleon and Squirrel. For Squirrel seed-var, the C2-GMAE values are averaged over the three fixed-mask settings in the appendix.
>
> | Setting | Chameleon C2-GMAE | Chameleon GraphMAE | Squirrel C2-GMAE | Squirrel GraphMAE |
> |---|---:|---:|---:|---:|
> | Seed-var, k=100 | 0.497 | 0.497 | 0.550 | 0.491 |
> | Seed-var, k=500 | 0.527 | 0.479 | 0.519 | 0.348 |
> | Seed-var, k=1000 | 0.503 | 0.460 | 0.493 | 0.346 |
> | Mask-var, k=100 | 0.199 | 0.166 | 0.184 | 0.179 |
> | Mask-var, k=500 | 0.195 | 0.159 | 0.182 | 0.143 |
> | Mask-var, k=1000 | 0.195 | 0.162 | 0.187 | 0.140 |
>
> These results show that C2-GMAE gives more stable reconstruction context than GraphMAE on both datasets.
>
> For "density", we define it as the average feature similarity between a node and its nearby neighbors because neighbors with more similar features usually provide more stable and useful information for reconstructing a masked node. So a higher score means an easier region for reconstruction, while a lower score means a harder region with weaker support.
>
> We use 2-hop neighbors because this gives a better balance between locality and stability. Using only 1-hop neighbors is too local and can be strongly affected by immediate neighborhood noise. Using 2-hop neighbors adds slightly broader local context and gives a more stable estimate of reconstruction support. In contrast, using 3-hop or 4-hop neighbors includes more distant and less relevant nodes, making the score less useful for separating easy and hard regions. This is also supported by our hop ablation:
>
> | Hop | Actor | Chameleon |
> |---|---:|---:|
> | 1-hop | 0.3982 ± 0.0100 | 0.8162 ± 0.0083 |
> | 2-hop | **0.4259 ± 0.0112** | **0.8242 ± 0.0071** |
> | 3-hop | 0.4165 ± 0.0174 | 0.8216 ± 0.0068 |
> | 4-hop | 0.4136 ± 0.0122 | 0.8205 ± 0.0092 |
>
> For "partitions", we sort nodes by density and split them into equal-size groups, so masking can be allocated more evenly across easy and hard regions.
>
> For "heterophily", it is defined by feature similarity between a node and its neighbors. We rank the neighboring edges by the feature similarity, and use the bottom-*p* edges with the lowest similarity as the heterophilic edges.
>
> **Response to Weakness 4:**
>
> Our experiments already follow a fixed setting across all methods. For node classification, all methods use the same GAT encoder with 4 heads and 1024 hidden units, using either 2 or 3 layers, with 2-layer MLP feature and position decoders, and 10 runs. For graph-level tasks, we use a 2-layer GatedGCN encoder, 2-layer MLP decoders, public OGB splits, and 10 runs. In the final version, we will provide a clearer implementation summary and release the code.

---

> > ### Author Rebuttal · Reviewer_Jv2E · 2026-04-01
> >
> > I thank the authors for their response. The rebuttal has addressed some of my concerns. However:
> > - Although responses are received, I cannot confirm whether the unclear presentation (Weakness 1) in the original paper has actually been resolved, especially given that multiple reviewers have also raised concerns.
> > - The authors' response has not alleviated my concerns regarding the motivation (Weakness 2). In their claim that "similar reconstruction loss does not always mean similar reconstruction context", the two aspects are not causally connected in the first place, as **the reconstruction context naturally varies across different training processes, so there is no need to select and compare checkpoints with similar reconstruction losses for the claim**. Moreover, the rationale behind such checkpoint selection remains unclear. Different training processes involve initializations, random masks, and even mini-batch orders. If the loss curves are not similar in shape, checkpoints with similar loss values map to different locations in the parameter space. Then, what are the connections between the checkpoints beyond the numerical coincidence of loss values? Why can they be compared? If, on the other hand, the loss curves are similar in shape, this would instead suggest that variations in reconstruction context do not have a notable impact on training in GMAEs.
> > - The methodology remains unclear and lacks necessary discussion (Weakness 3). For instance, the term "density partition" may cause confusion with existing concepts such as (edge-based) graph density and density-based partitioning methods like DBSCAN. To my knowledge, average feature similarity does not inherently represent "density" either. Furthermore, if nodes are simply split into equal-sized groups, this appears to be a heuristic sampling strategy, as the masked probability distribution of a certain heterophilic node remains unchanged (uniform). To me, a more intuitive and effective choice could be adaptively increasing the masked probability for heterophilic nodes.
> >
> > Based on the current situation, despite my appreciation of the authors' efforts, I believe this paper still does not reach the expected quality of the ICML conference. I will maintain my score.

---

> > > ### Author Response · Authors · 2026-04-07
> > >
> > > Thank you for your comments.
> > >
> > > **Response to Concern 1**
> > >
> > > During the discussion, we can’t revise the original PDF, so the rebuttal can only clarify the intended meanings. In the revision, we will revise terms where they first appear, define “reconstruction context” as the neighborhood information and weighting pattern for reconstructing a masked node, explain “role-mixed neighborhoods,” “boundary/conflict cues,” “hard regions,” and “limited coverage” plainly, rewrite the caption of Fig. 1, and revise Fig. 2 to define “reconstruction-matched checkpoints,” explain how “reconstruction context” is measured, clarify axes and rising curves, state that Fig. 2 is a diagnostic study, and move key protocol details from Appendix to the main text.
> > >
> > > **Response to Concern 2**
> > > - Reconstruction context is the source from which the representation of a masked node is learned, and it influences reconstruction loss. Because reconstruction context naturally varies across training processes, this motivates us to examine how it is used and how to use it better. Different neighbors and weights can change representation quality. Thus, even when reconstruction losses are similar, differences in reconstruction context usage can produce representations of varying quality, which in turn lead to different downstream performance. Such differences can't be directly reflected by scalar reconstruction loss. What we expect is stable context usage across training processes and high-quality representations.
> > >
> > > - The checkpoint protocol is used only in diagnostic analyses as a control for comparable reconstruction levels. The rationale is to compare runs under a common late-stage reconstruction regime within the same model–dataset setting, rather than arbitrary checkpoints with numerically similar loss values. For each model–dataset setting, we run a preliminary training process, identify the stage where training reconstruction loss becomes stable, and choose a representative target loss value from this regime. For each run under that setting, we select the checkpoint whose training reconstruction loss is closest to that target value. Thus, within one model–dataset setting, the target value is fixed across runs, so the selection is not based on numerical coincidence. For different models or datasets, the target value is chosen separately because the scale of reconstruction loss is setting-dependent. This protocol does not depend on loss-curve shape. Different runs may have different loss-curve shapes, but as long as they reach the same stable late-stage regime, the selected checkpoints are compared under the same reconstruction condition. It is therefore a diagnostic control, not an assumption of parameter-space proximity.
> > >
> > > **Response to Concern 3**
> > > - Conceptually, density is a general metric in machine learning to characterize the concentration of data points. One metric can be regarded as density if it satisfies the above criterion, even though the density used in DBSCAN is one of the most representative metrics. To name a few, [1] defines density through the entropy of the feature similarity; [2] defines it through the squared $L_2$-norm of a neural network. More density definitions can be found in [3–5]. In our work, density characterizes the concentration of local reconstruction support. We use average feature similarity between a node and its 1-hop and 2-hop neighbors, because neighbors with higher feature similarity may provide more coherent support for masked-node reconstruction.
> > >
> > > [1] Tune it the Right Way: Unsupervised Validation of Domain Adaptation via Soft Neighborhood Density. ICCV 2021.
> > >
> > > [2] Squared Neural Families: A New Class of Tractable Density Models. NeurIPS 2023.
> > >
> > > [3] A Domain Adaptive Density Clustering Algorithm for Data with Varying Density Distribution. TKDE 2019.
> > >
> > > [4] Nonparametric Density Estimation under Distribution Drift. ICML 2023.
> > >
> > > [5] Conditional Density Estimation with Histogram Trees. NeurIPS 2024.
> > >
> > > - We agree that our density-partitioned masking is a heuristic sampling strategy. Its purpose is to stratify nodes by local support level, so each training round covers multiple support levels instead of being dominated by one type of region. To verify whether it is more effective to increase the probability of masking heterophilic nodes, we also conducted an experiment as follows. Nodes are divided into $B$ partitions by average feature similarity. The $k$-th partition is assigned weight $w_k = 1 + \frac{k-1}{B-1}(b-1)$, and its masking quota is $q_k = \frac{w_k}{\sum_{j=1}^{B} w_j} M$. A higher $b$ indicates a higher probability for heterophilic nodes.
> > >
> > > | Dataset | Ours | $b=2.0$ | $b=3.0$ |
> > > |---|---:|---:|---:|
> > > | Chameleon | 82.42$\pm$0.71 | 81.50$\pm$1.11 | 81.10$\pm$1.00 |
> > > | Actor | 42.59$\pm$1.12 | 41.08$\pm$1.02 | 40.89$\pm$2.27 |
> > >
> > > Our strategy performs best, which suggests that increasing the masking probability of heterophilic nodes is not a better choice.

---

### Official Review · Reviewer_QY8r · 2026-03-03

**Soundness:** 4
**Presentation:** 4
**Significance:** 3
**Originality:** 3
**Overall Recommendation:** 5
**Confidence:** 5

**Summary:**

This paper studies Graph Masked Auto Encoder training and points out a simple issue. Two models can reach similar reconstruction loss but use different neighbors as reconstruction context, and this can change downstream accuracy. The authors propose C2-GMAE to encourage more consistent context use and better training coverage. The method uses three main ideas: relative positional distances from Laplacian eigenvectors to guide edge scoring, density partitioned masking to cover sparse regions, and heterophilic edge amplification to keep useful cross-type signals during reconstruction. Experiments on eight node classification graphs and six OGB graph prediction datasets show strong results, and the ablations support the value of the main components.

**Compliance With Llm Reviewing Policy:**

Affirmed.

**Key Questions For Authors:**

1. For density partitioned masking, can you briefly explain why you use feature similarity to define density in this setting? If the intuition is clearly stated, I would judge the design choice as more justified and would improve my presentation assessment.

2. What is the preprocessing time and memory for computing Laplacian eigenvectors on your largest graphs, and how does it compare with training time? If the cost is modest or can be amortized well, it would reduce my concern about practicality and would strengthen my overall score.

3. Could you provide a brief intuition on why the global Top-$k$ overlap in Eq. 21 was chosen as the primary metric for the claim verification in the appendix, rather than sticking exclusively with the per-node Top-$m$ overlap used in Figure 2?
If the authors provide a clear justification for using the global Top-$k$ overlap, I would be more confident that the diagnostic study supports the paper’s main claims, and this would improve my soundness assessment.

4. In Table 3, the ablations show clear performance drops when removing heterophilic amplification, Variant B, on heterophilic graphs. For the homophilic graphs evaluated, does this heterophilic amplification component have a neutral or slightly negative effect? If the authors clarify the effect of this module on homophilic graphs, I would have a clearer view of when the component is most needed and this would strengthen my significance assessment.

**Limitations:**

This manuscript currently lacks a dedicated discussion of its technical limitations. It would be highly beneficial to explicitly acknowledge the potential computational bottleneck of preprocessing Laplacian eigenvectors for massive graphs. Additionally, discussing the boundary conditions of the proposed method, such as scenarios where heterophilic amplification might not yield benefits (e.g., on strictly homophilic datasets), would provide a more comprehensive view of the framework. Including a short limitations section addressing these points would strengthen the paper.

**Strengths And Weaknesses:**

Strengths
1. Clear motivation backed by a controlled study that links context differences to accuracy shifts.

2 . The design matches the goal. Positional distances guide attention, density partitioned masking improves coverage, and heterophilic amplification targets role mixed neighborhoods.

3. Strong results on both node-level and graph-level benchmarks, with broad baseline coverage and solid ablations.

Weaknesses
1. The paper should explain more clearly why feature similarity is used to define density for the masking bins.

2. The paper relies on Laplacian eigenvectors, and the preprocessing cost is not clearly reported in the main text.

3. Some notations and concepts are not clearly described. Please refer to Key Questions For Authors below.

---

> ### Author Rebuttal · Authors · 2026-03-31
>
> Dear Reviewer QY8r:
>
> We are very grateful for your valuable comments and questions. The responses are as follows:
>
> **Response to Weakness 1:**
>
>  We agree that the intuition behind the density definition should be stated more clearly. Our density is not structural density; it measures local reconstruction support for a masked node. Please see **Response to Question 1**.
>
> **Response to Weakness 2:**
>
> We agree that the practicality of the Laplacian-based positional encoding should be explained more clearly. In the revision, we will state explicitly that this is a one-time offline preprocessing step and add a concise complexity discussion. Please see **Response to Question 2**.
>
> **Response to Weakness 3:**
>
> We agree that some notations and concepts can be presented more clearly. In the revision, we will clarify: (1) the meaning of density; (2) the roles of per-node Top-$m$ and global Top-$k$; and (3) when heterophilic amplification (HA) is most useful. Please see **Response to Questions 1, 3, and 4**.
>
> **Response to Question 1:**
>
> We define density as the average feature similarity between a node and its nearby neighbors because neighbors with more similar features usually provide more stable and useful evidence for reconstructing a masked node. Thus, a higher score indicates an easier region for reconstruction, while a lower score indicates a harder region with weaker support.
>
> We use 2-hop neighbors because this gives a better balance between locality and stability. Using only 1-hop neighbors is often too local and sensitive to immediate neighborhood noise. Adding 2-hop neighbors introduces slightly broader local context and yields a more stable estimate of reconstruction support. In contrast, 3-hop or 4-hop neighbors introduce more distant and less relevant nodes, making the score less effective for separating easy and hard regions. For more details, please refer to **Response to Weakness 3 of Reviewer Jv2E**.
>
> **Response to Question 2:**
>
> Let $N$ be the number of nodes, $|E|$ the number of edges, $K_u$ the positional dimension, and $L$ the number of layers. For sparse graphs, truncated Laplacian eigenvectors are precomputed once per graph with approximate cost $O(rK_u(|E|+N))$, where $r$ is the number of solver iterations. During training, the added positional computation is sparse and performed on $E_{pe}$, with cost $O((|E|+N)K_u)$; the heterophily indicator $H$ can also be precomputed. The encoder still performs sparse message passing / attention over $L$ layers, so the overall overhead is mainly a constant-factor increase over the backbone. The positional loss is computed only on $\Omega$, costing $O(|\Omega|)$ with $|\Omega|\le |E|$. Extra memory mainly comes from $U$ in $O(NK_u)$, optional edge-wise $PE/H$ statistics in $O(|E|+N)$ if stored explicitly, and logits/targets on $\Omega$ in $O(|\Omega|)$. We will clarify this in the revision and note that spectral preprocessing may still be nontrivial on very large graphs.
>
> **Response to Question 3:**
>
> The two overlap metrics serve different purposes. The per-node Top-$m$ overlap in Fig. 2 is a local diagnostic showing whether a masked target relies on stable neighbors as reconstruction context. The global Top-$k$ overlap in Eq. (21) is a graph-level summary, which is more suitable for systematic comparison across runs and settings using a single scalar. We use the former for intuition and visualization, and the latter for compact claim verification. We will clarify this distinction in the final version.
>
> **Response to Question 4:**
>
> Our ablations focus on heterophilic datasets because HA is designed to preserve informative cross-type signals in role-mixed neighborhoods. To directly answer your question, we additionally tested removing HA on the two homophilic datasets.
>
> | Dataset | Full C2-GMAE | w/o HA  |
> |---|---:|---:|
> | Facebook | 91.57±0.16 | 91.288±0.161  |
> | WikiCS | 79.59±0.15 | 78.875±0.153  |
>
> These results show that HA brings a modest but consistent gain on homophilic graphs, while its effect is smaller than on heterophilic graphs. This matches our intuition that HA is most useful when cross-type relational signals are informative.
>
> **Response to Limitations:**
> We agree that our manuscript lacks a limitations section. We will clarify that preprocessing truncated Laplacian eigenvectors may become a computational bottleneck on massive graphs, although this cost is paid only once in offline preprocessing. We will also state that the benefit of heterophilic-edge amplification depends on graph characteristics, may be smaller on strictly or strongly homophilic datasets, and may vary under more extreme heterophilic settings. In addition, our method introduces extra hyperparameters and incurs additional time and memory overhead compared with simpler GMAE baselines. We will add a limitation section in final version, for more details please see **Response to Limitations of Reviewer E6kc.**

---

> > ### Author Rebuttal · Reviewer_QY8r · 2026-04-01
> >
> > Thanks for the detailed rebuttal, which has addressed my concerns, and thus I have raised the scores of Soundness and Presentation, and I would like recommend accepting this paper.

---

> > > ### Author Response · Authors · 2026-04-07
> > >
> > > We sincerely thank you for your positive feedback, for raising your scores. We are also very glad that our rebuttal addressed your concerns.

---

### Official Review · Reviewer_E6kc · 2026-03-11

**Soundness:** 2
**Presentation:** 3
**Significance:** 3
**Originality:** 3
**Overall Recommendation:** 4
**Confidence:** 4

**Summary:**

This paper studies an interesting issue in graph masked auto-encoders: similar reconstruction outcomes can arise from different ways of using neighborhood context, which may lead to unstable downstream performance. It proposes C2-GMAE, a consistency- and coverage-aware framework that anchors context scoring with positional structure, improves masking coverage across graph regions, and strengthens heterophilic relational cues during reconstruction. Experiments on both node-level and graph-level benchmarks show improved performance over strong GMAE baselines.

**Compliance With Llm Reviewing Policy:**

Affirmed.

**Final Justification:**

The authors addressed some of my concerns, so I am willing to raise my score. However, some of my other concerns still remain, such as the assumption that “Laplacian eigenvectors $U$ are available through preprocessing,” since this is unrealistic for large-scale graphs in practice.

**Key Questions For Authors:**

1. The motivation that reconstruction loss is an imperfect proxy for representation quality has been explored in prior work, but this connection is not discussed in the paper. For example, SDMG [1] studies reconstruction–representation misalignment from a frequency perspective, while Joint-Embedding vs Reconstruction [2] provides a theoretical comparison between input-space reconstruction and latent-space alignment.

[1] *SDMG: Smoothing Your Diffusion Models for Powerful Graph Representation Learning.* ICML 2025.

[2] *Joint-Embedding vs Reconstruction: Provable Benefits of Latent Space Prediction for Self-Supervised Learning.* NeurIPS 2025.

2. I checked both the main text and the appendix, and it seems that the diagnostics for Claim0, Claim1, and Claim2 are all conducted almost entirely on the Squirrel dataset. Do the same phenomena and conclusions also hold for other graphs, such as Chameleon? Validating these claims on a single dataset is not sufficiently convincing.

3. If I understand correctly, does the method assume from the outset that the Laplacian eigenvectors U are already available through preprocessing? This could be quite expensive on large graphs.

4. The paper defines “reconstruction context” using the Top-m/Top-k neighbors or edges ranked by decoder attention. Does this attention overlap truly reflect the context the model relies on for reconstruction, or is it merely a correlated proxy?

**Limitations:**

The paper does not include a limitations section.

**Strengths And Weaknesses:**

**Strengths:**

1. The paper introduces a meaningful perspective on the limitations of reconstruction-based objectives.
2. It moves the focus from reconstruction outcomes to context usage during reconstruction, which is a clear angle.
3. The evaluation covers both node-level and graph-level benchmarks, rather than validating the method on only a single task.

**Weaknesses:**

1. Several of the paper's main claims are validated only on a single dataset, which limits the strength and generality of the conclusions.
2. The method relies on precomputed Laplacian eigenvectors.
3. The method appears to require nontrivial tuning, and the preferred hyperparameter ranges vary noticeably across datasets. For example, the hyperparameters reported in Table 10 differ substantially from one dataset to another, and Figures 4 and 5 also suggest inconsistent preferred ranges.

---

> ### Author Rebuttal · Authors · 2026-03-31
>
> Dear Reviewer E6kc:
>
> Thank you for valuable comments. Our responses are below:
>
> **Response to Weakness 1:**
> We agree that validating the claim-based diagnostics on only one dataset is insufficient. We reran Claim0/1/2 on Chameleon and observed the same overall trends. Please see **Response to Question 2** for details.
>
> **Response to Weakness 2:**
> We agree that precomputed truncated Laplacian eigenvectors add preprocessing cost, especially on large graphs. In our implementation, however, this is a one-time offline step with no extra training cost. Please see **Response to Question 3** for details.
>
> **Response to Weakness 3:**
> In Figs. 4-5, the y-axis is zoomed in, so the fluctuations appear larger than they are. The absolute performance differences are generally small, suggesting stability within a limited search range. Although the preferred values differ somewhat across datasets, we view this as normal dataset-dependent tuning rather than instability.
>
> **Response to Question 1:**
> We agree that this connection should be discussed more clearly. SDMG studies reconstruction-representation misalignment from a frequency perspective, while Joint-Embedding vs Reconstruction compares input-space reconstruction with latent-space alignment. Our motivation is related, but our focus is different. We study a process-level issue in graph masked auto-encoding: even when reconstruction outcomes are similar, the model may rely on very different neighborhood context across seeds or masks, leading to different downstream performance. We further observe that this issue may be more severe in harder graph regions, where standard masking provides weaker training pressure. We will add this discussion in final version.
>
> **Response to Question 2:**
> We agree that the original appendix evidence was only on Squirrel. We therefore reran Claim0/1/2 on Chameleon, and the same overall conclusions still hold.
>
> **Claim0/1: context-overlap stability (higher is better)**
>
> | Setting | Full | Noanchor | Nohint |
> |---|---:|---:|---:|
> | Seed-var, \($J_k$\) @100/500/1000 | 0.497 / 0.527 / 0.503 | 0.448 / 0.456 / 0.489 | 0.392 / 0.487 / 0.496 |
> | Mask-var, \($J_k$\) @100/500/1000 | 0.199 / 0.195 / 0.195 | 0.095 / 0.094 / 0.092 | 0.192 / 0.191 / 0.192 |
>
> **Claim2: coverage under density partitions**
>
> | Bins | Metric | Full | Random |
> |---|---|---:|---:|
> | Real, \(k=1000\) | Entropy / Std | 3.601 / 0.0108 | 3.597 / 0.0113 |
> | Real, \(k=500\) | Entropy / Std | 3.582 / 0.0118 | 3.579 / 0.0118 |
> | Shuffled, \(k=1000\) | Entropy / Std | 3.673 / 0.0045 | 3.675 / 0.0041 |
> | Shuffled, \(k=500\) | Entropy / Std | 3.667 / 0.0052 | 3.667 / 0.0051 |
>
> These results support the same overall conclusions beyond Squirrel. In particular, the full model shows better context-overlap stability than Noanchor under both Seed-var and Mask-var. For Claim2, the advantage appears under real density bins but disappears after bin shuffling, suggesting that the effect comes from meaningful density partitions rather than random grouping. We will add these results.
>
> **Response to Question 3:**
> Yes. Our method assumes that truncated Laplacian eigenvectors $U$ are available through preprocessing. In our implementation, we compute $U$ following standard practice [1]. We agree that this step can be costly on very large graphs. However, it is a one-time offline step: $U$ is computed once, saved, and then directly loaded during training. We also use a truncated eigenspace instead of the full spectrum.
>
> [1] Graph Positional Autoencoders as Self-supervised Learners. KDD 2025.
>
> **Response to Question 4:**
> We agree that decoder-attention overlap is not a complete causal explanation of reconstruction context. In our framework, reconstruction is directly determined by edge-level attention scores, and C2-GMAE changes this scoring through heterophilic-edge amplification. Therefore, attention overlap can reflect which context edges the model relies on more during reconstruction, so we use it as a practical proxy.
>
> **Response to Limitations:**
> We agree that the current manuscript lacks a limitations section. The limitations of our method are as follows. First, preprocessing truncated Laplacian eigenvectors can become a computational bottleneck on massive graphs, although this cost is paid only once in offline preprocessing rather than during training. Second, the effect of heterophilic-edge amplification depends on graph characteristics and may not always be beneficial; for example, its gain may be smaller on strictly or strongly homophilic graphs, and performance under more extreme heterophilic settings may also vary. Third, our method introduces additional hyperparameters, which makes model selection and tuning more complex. Finally, compared with simpler GMAE baselines, our method also incurs extra time and memory overhead due to its additional components. We will add this limitations section in the final version.

---

> > ### Author Rebuttal · Reviewer_E6kc · 2026-04-03
> >
> > The authors addressed some of my concerns. I believe my questions can be addressed in the camera-ready version, so I am willing to raise my score now. However, I am still not satisfied with the assumption that “Laplacian eigenvectors *U* are available through preprocessing.” Although this is a one-time cost, for even moderately large graphs, the preprocessing may take several days, which could be far more time-consuming than the training itself.

---

> > > ### Author Response · Authors · 2026-04-07
> > >
> > > We sincerely thank you for your positive feedback and we are very glad that our rebuttal addressed your main concerns.
> > >
> > > Regarding the remaining concern on preprocessing Laplacian eigenvectors $U$, we agree that this is a limitation of the current method. We will discuss this limitation more clearly in the final version and state the applicable setting of our method more explicitly.

---

### Decision · Program_Chairs · 2026-04-30

**Decision:**

Accept (regular)

**Comment:**

This paper studies an interesting issue in graph masked auto-encoders: similar reconstruction outcomes can arise from different ways of using neighborhood context, which may lead to unstable downstream performance. It proposes C2-GMAE, a consistency- and coverage-aware framework that anchors context scoring with positional structure, improves masking coverage across graph regions, and strengthens heterophilic relational cues during reconstruction. The authors response to all issues raised by the reviewers. From my view, the paper is in a good quality. I would recommend the authors carefully revise the paper according to the comments in final submission.